# Think-on-Graph 2.0: Deep and Faithful Large Language Model Reasoning with Knowledge-guided Retrieval Augmented Generation

**Shengjie Ma**[1,2*], **Chengjin Xu**[1*†], **Xuhui Jiang**[1*], **Muzhi Li**[3], **Huaren Qu**[4],
**Cehao Yang**[1], **Jiaxin Mao**[2†], **Jian Guo**[1†]

[1] IDEA Research, International Digital Economy Academy, Shenzhen, Guangdong, China
[2] Gaoling School of Artificial Intelligence, Renmin University of China, Beijing, China
[3] The Chinese University of Hong Kong, Sha Tin, New Territories, Hong Kong, China
[4] The Hong Kong University of Science and Technology, Guangzhou, Guangdong, China

## Abstract

Retrieval-augmented generation (RAG) has improved large language models (LLMs) by using knowledge retrieval to overcome knowledge deficiencies. However, current RAG methods often fall short of ensuring the depth and completeness of retrieved information, which is necessary for complex reasoning tasks. In this work, we introduce Think-on-Graph 2.0 (ToG-2), a hybrid RAG framework that iteratively retrieves information from both unstructured and structured knowledge sources in a tight-coupling manner. Specifically, ToG-2 leverages knowledge graphs (KGs) to link documents via entities, facilitating deep and knowledge-guided context retrieval. Simultaneously, it utilizes documents as entity contexts to achieve precise and efficient graph retrieval. ToG-2 alternates between graph retrieval and context retrieval to search for in-depth clues relevant to the question, enabling LLMs to generate answers. We conduct a series of well-designed experiments to highlight the following advantages of ToG-2: 1) ToG-2 tightly couples the processes of context retrieval and graph retrieval, deepening context retrieval via the KG while enabling reliable graph retrieval based on contexts; 2) it achieves deep and faithful reasoning in LLMs through an iterative knowledge retrieval process of collaboration between contexts and the KG; and 3) ToG-2 is training-free and plug-and-play compatible with various LLMs. Extensive experiments demonstrate that ToG-2 achieves overall state-of-the-art (SOTA) performance on 6 out of 7 knowledge-intensive datasets with GPT-3.5, and can elevate the performance of smaller models (e.g., LLAMA-2-13B) to the level of GPT-3.5's direct reasoning. The source code is available on `https://github.com/IDEA-FinAI/ToG-2`.

## 1 Introduction

Retrieval-augmented generation (RAG) has significantly enhanced the capabilities of large language models (LLMs) by retrieving relevant knowledge from external sources, as a promising solution to address the knowledge deficiencies and hallucination issues (Zhao et al., 2024). Despite this potential, and researchers' attempts to incorporate various complex additional processes into RAG (Yu et al., 2023; Edge et al., 2024; Li et al., 2024c), LLMs still struggle to maintain human-like reasoning trajectories (Kahneman, 2011) when handling complex tasks, which require continuous effort and motivation to integrate fragmented information and the structural relationships between them.

Many recent implementations of RAG mainly rely on vector retrieval of textual content within documents (Ding et al., 2024; Asai et al., 2023; Wang et al., 2023b). Although vector embedding is effective for measuring the semantic similarity between two texts, text-based RAG struggles to capture the structured relationship between texts and documents (Chen et al., 2023). Specifically, simple

---

[*]Both authors contributed equally to this research.
[†]Corresponding author

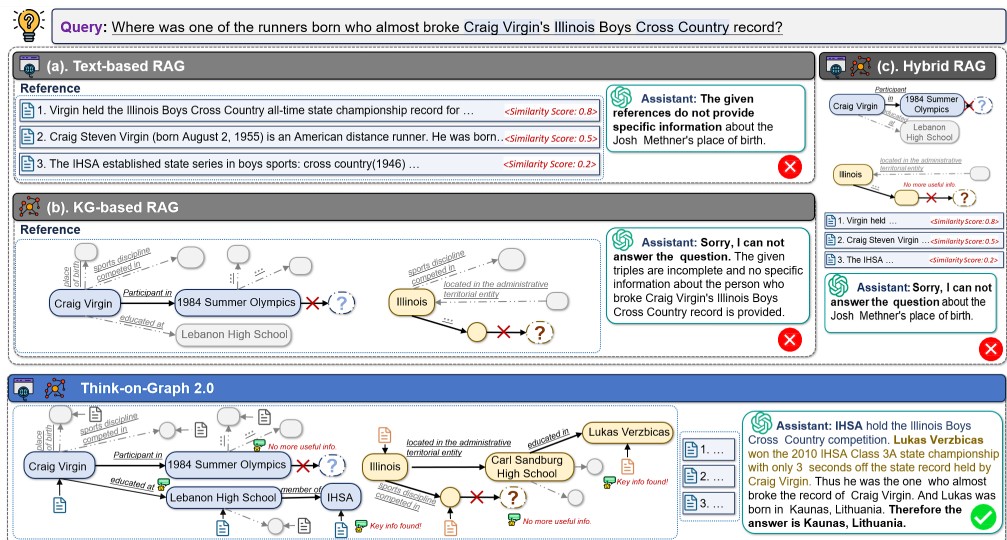

Figure 1: The illustration of differences among (a) text-based RAG, (b) KG-based RAG, (c) KG+Text (Loose-coupling) Hybrid RAG, and (d) our proposed KG×Text (Tight-coupling) RAG framework.

vector-based matching may overlook knowledge-level association between entities within different texts, e.g., *Global Financial Crisis* and *The 2008 Recession*, which are two surface names that refer to the same event; *Harry Potter* and *Fantastic Beasts* which are both written by *J.K. Rowling*. This limitation results in that most text-based RAG approaches are not suitable for multi-step reasoning or tracking logical links between different information fragments. As shown in Figure 1(a), the naive RAG, where generic retrievers search through large-scale corpora, the recall often focuses on the level of superficial semantic similarity (Edge et al., 2024), but overlooks the structured relationship between *Craig Virgin* and *Lukas Verzbicas*, as well as texts containing one of them.

Knowledge Graphs (KGs) organize the structural relationships between entities from fragmented information and store knowledge in the form of triples. Some researchers have explored KG-based RAG approaches (Sun et al., 2024), which prompt LLMs by retrieving KG triples relevant to the question. Although KGs are effective for structuring high-level concepts and relationships, they inherently suffer from inner incompleteness and a lack of information beyond their ontology Li et al. (2024a;b). As shown in Figure 1(b), the KG is unable to provide detailed information about *Lukas Verzbicas*'s competition records. Some recent works focus on integrating text-based and KG-based RAG systems (Li et al., 2024c; Edge et al., 2024). In this approach, information retrieved from both structured and unstructured knowledge sources is aggregated to prompt the LLM to answer questions but does not improve the retrieval results on one knowledge source through the other. As shown in Figure 1(c), such loose-coupling combination (KG+Text) approaches still fall short in handling complex queries that require detailed information returned obtained through in-depth retrieval.

Considering the aforementioned challenges, we propose **T**hink-**o**n-**G**raph **2.0** (ToG-2), a tight-coupling hybrid (KG×Text) RAG paradigm which effectively integrates unstructured knowledge from texts with structured insights from KGs, serving as a roadmap to enhance complex problem-solving. As shown in Figure 1(d), ToG-2 first initializes the starting point for graph search by extracting topic entities from the question. At the beginning of each retrieval round, ToG-2 explores more candidate entities by performing relation retrieval on the KG and retrieves relevant text from documents associated with these candidate entities. Following knowledge-guided context retrieval, ToG-2 prunes the candidate entities based on the results of the context retrieval, updates the set of topic entities and uses it as the starting point for the next round of retrieval. After each retrieval round, ToG-2 prompts the LLM to answer the question based on the highly relevant knowledge triples and contexts obtained during the retrieval process. If the retrieved information is insufficient to answer the question, the next round of retrieval continues for searching more in-depth clues. Such a tight-coupling hybrid RAG paradigm makes LLM perform closer to human when solving complex

problems: examining current clues and associating potential entities based on their existing knowledge framework, continuously digging into the topic until finding the answer.

The advantage of ToG-2 can be summarized as: (1) **In-depth retrieval:** ToG-2 achieves in-depth and reliable context retrieval through the guide of KGs and performs precise graph retrieval by treating documents as node contexts, achieving a tight-coupling combination of KG and text RAG, enabling deep and comprehensive retrieval processes. (2) **Faithful reasoning:** ToG-2 iteratively performs a collaborative retrieval process based on KG and text, using retrieved heterogeneous knowledge as the basis for LLM reasoning and enhancing the faithfulness of LLM-generated content (3) **Efficiency and Effectiveness:** a) ToG-2 is a training-free and plug-and-play framework that can be applied to various LLMs; b) ToG-2 can be executed between any associated KG and document database, while for purely document database, entities can be extracted from the documents first, and then a graph can be constructed through relation extraction or entity co-occurrence (Edge et al., 2024); c) ToG-2 achieves new SOTA performances on various complex knowledge reasoning datasets and can elevate the reasoning capabilities of smaller LLMs, e.g., Llama2-13B to a level comparable to direct reasoning with powerful LLMs like GPT-3.5.

## 2 RELATED WORKS

RAG aims to offer effective utilization of external knowledge sources with high interpretability. An important factor is the type of data retrieved. In this section, we will briefly review text-based RAG, KG-based RAG and hybrid RAG.

Text-based RAG approaches retrieve information based on the semantic similarities between questions and texts (Gao et al., 2024; Qiu et al., 2024). However, these approaches have difficulties in capturing in-depth relationships between texts and the retrieved texts may contain redundant content. To address these challenges, ITER-RETGEN (Shao et al., 2023) follows an iterative strategy, merging retrieval and generation in a loop, alternating between "retrieval-augmented generation" and "generation-augmented retrieval". Trivedi et al. (2023) combined RAG with the Chain of Thought (CoT) (Wei et al., 2022a) method, alternating between CoT-guided retrieval and retrieval-supported CoT processes, significantly improving GPT-3's performance on various Q&A tasks. While these optimizations enhance the recall of relevant texts, the whole retrieval process becomes more time-consuming and errors may accumulate over iterations, due to the lack of a reliable guide.

The structured knowledge representation of KGs is particularly beneficial to LLMs because it introduces a level of interpretability and precision in the knowledge. Early approaches (Sun et al., 2020; Peters et al., 2019; Huang et al., 2024; Liu et al., 2020) focused on embedding knowledge from KGs directly into the neural networks of LLMs. To maximize the utilization of the interpretability of the KG, more recent studies (Jiang et al., 2024; Sun et al., 2024) have shifted towards using KGs to augment LLMs externally rather than embedding the knowledge directly into the models. KG-based RAG approaches involve translating relevant structured knowledge from KGs into textual prompts that are then fed to LLMs, however naturally suffer from the knowledge incompleteness of KGs.

The most relevant works to ToG-2 are hybrid RAG approaches that aim to integrate both KGs and unstructured data with LLM, leveraging the strengths of both to mitigate their respective weaknesses. Chain-of-Knowledge (Li et al., 2024c) (CoK) is a hybrid RAG method that retrieves knowledge from Wikipedia, Wikidata and Wikitable to ground LLMs' outputs. GraphRAG (Edge et al., 2024) establishes a KG from documents and enables KG-enhanced text retrieval. HybridRAG (Sarmah et al., 2024) retrieves information from both vector databases and KGs, achieving superior reasoning performance compared to either text-based RAG or KG-based RAG methods alone. However, existing hybrid RAG approaches merely aggregate information retrieved from KGs and texts but do not improve the retrieval results on one knowledge source through the other. In this work, ToG-2 aims to tightly couple KG-based RAG and text-based RAG methods, enabling both in-depth context retrieval and precise graph retrieval to enhance complex reasoning performances of LLMs.

## 3 METHODOLOGY

The proposed method ToG-2 draws from the ToG approach Sun et al. (2024) in multi-hop searches within KGs, starting from key entities identified in the query and exploring outward based on en-

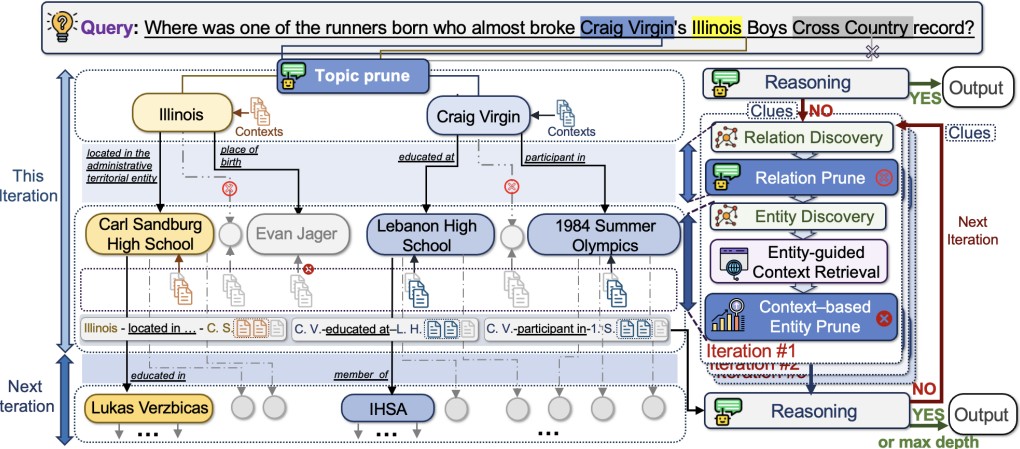

Figure 2: An example illustrating the workflow of ToG-2.

tities and relationships with a prompt-driven inferential process. ToG-2 combines the logical chain extensions based on triples with unstructured contextual knowledge of relevant entities by iteratively performing knowledge-guided context retrieval and context-enhance graph retrieval, thus more effectively integrating and utilizing heterogeneous external knowledge.

Specifically, ToG-2 begins by extracting entities from the given question as initial topic entities. It then performs an iterative process of graph retrieval, context retrieval and LLM reasoning. At the start of each iteration, ToG-2 selectively explores entities neighboring the current topic entities on the KG, using the newly encountered entities to refine the retrieval scope, thereby enhancing both efficiency and accuracy. Next, ToG-2 ranks and selects entities based on the query and the contextual knowledge retrieved from relevant documents, reducing ambiguity and ensuring accurate exploration for the next step. After, the LLM utilizes heterogeneous knowledge, including triple paths and entity contexts, to either answer the question or proceed with further rounds of retrieval if the information gathered is insufficient. In this section, we will explain each step in detail.

## 3.1 INITIALIZTION

Given a question $q$, ToG-2 first identifies the entities present in $q$ and links them to entities in the KG. This step can be done by different entity linking (EL) methods, such as LLMs or specialized EL tools [1]. Then, ToG-2 performs a Topic Prune (TP) step to select suitable entities as starting points of exploring in a KG, which prompts the LLM to evaluate $q$ and appearing entities, selecting topic entities $\mathcal{E}^0_{topic} = \{e_1, e_2, \ldots, e_N\}$ and $N$ is determined by LLM.

Prior to the 1st-round graph retrieval, ToG-2 uses dense retrieval models (DRMs, including dual-tower and single-tower models) to extract the top-$k$ chunks from the documents associated with the initial topic entities $\mathcal{E}^0_{topic}$. The LLM then evaluates whether this information is sufficient to answer the question, drawing on its own knowledge. If it concludes that the available information is adequate, no further steps will be required.

## 3.2 HYBRID KNOWLEDGE EXPLORATION

The following will illustrate how ToG-2 iteratively harmonizes and tightly couples heterogeneous knowledge. The iterative process will be explained in two main parts, **Context-enhanced Graph Search** and **Knowledge-guided Context Retrieval**. Formally, in the $i$-th iteration, the topic entities are denoted as $\mathcal{E}^i_{topic} = \{e^i_1, e^i_2, \ldots, e^i_j\}$ and their preceding triple paths are $\mathcal{P}^{i-1} = \{P^{i-1}_1, P^{i-1}_2, \ldots, P^{i-1}_j\}$, $P^{i-1}_j = \{p^0_j, p^1_j, \ldots, p^{i-1}_j\}$, where $j \in [1, W]$, $W$ is a hyperparameter of exploration width (the max number of retained topic entities in each iteration, and $p^{i-1}_j$ is a

---

[1]We use entity linking API provided by Azure AI for Wikipedia, https://learn.microsoft.com/en-us/azure/ai-services/language-service/entity-linking/overview?source=recommendations

single triple $(e_j^{i-1}, r_j^{i-1}, e_j^i)$ containing $r_j^{i-1}$ as the relation between $e_j^{i-1}$ and $e_j^i$ in the KG, which could be either direction. Note that $i = 0$ indicates the initialization phase and the $P^0$ is empty.

### 3.2.1 KNOWLEDGE-GUIDED GRAPH SEARCH

By leveraging the rich structural connectivity of knowledge on the KG, the graph search aims to explore and establish high-level concepts and relationships between the question and target information that are seemingly distant from each other in semantic space.

**Relation Discovery:** At the beginning of the $i$-th iteration, through the function

$$\text{Edge}(e_j^i) = \{(r_{j,m}^i, h_m) \mid h \in \{\text{True}, \text{False}\}\}, \tag{1}$$

we can find the relations for all topic entities. $Edge()$ is a function that searches for relationships of entities. $h$ indicates whether the direction of that relation $r_{j,m}^i$ is pointing to the topic entity $e_j^i$.

**Relation Prune (RP):** From the collected relation set $\{Edge(e_j^i)\}_{j=1}^W$, We prompt the LLM to select and score the relations that are likely to find entities containing helpful context information for solving $q$. We design two prompting manners:

$$\text{PROMPT}_{RP}(e_j^i, q, Edge(e_j^i)), \tag{2}$$

$$\text{PROMPT}_{RP\_cmb}(\mathcal{E}_{topic}^i, q, \{Edge(e_j^i)\}_{j=1}^W). \tag{3}$$

The detailed prompt is shown in Appendix E Table 15, 16 and 17. The relations with low scores will be pruned. Equation 2 involves multiple calls to the LLM for entity pruning on each topic entity individually, while Equation 3 processes the relation selection for all topic entities in a single operation. Equation 2 simplifies the task for the LLM, but it is less efficient. Equation 3 reduces the number of API calls, thereby enhancing inference speed, and allowing the LLM to consider the interconnections between multiple reasoning paths simultaneously, facilitating selections from a broader perspective. However, when all topic entities recall an excessive number of relations, this may challenge the capacity of a weak LLM to effectively handle long texts. The selected relations for all topic entities in the $i$-th iteration are denoted as $\mathcal{R}^i = \{r_{j,m}^i\}$ ($j \in [1, W]$). In the case of Figure 2, the topic entity $e_1^0 = $ *Crag Virgin* and its relation $r_{1,1}^0 = $ *place_of_birth* is pruned since it may lead to a location that is unlikely to be relevant to a runner's performance.

**Entity Discovery:** Given a topic entity $e_j^i$ in $\mathcal{E}_{topic}^i$ and its corresponding selected relations $r_{j,m}^i$ in $\mathcal{R}^i$, function

$$\text{Tail}(e_j^i, (r_{j,m}^i, h_m)) = c_{j,m}^i) \tag{4}$$

identifies the connected entities $\{c_{j,m}^i\}$ of the topic entities $e_j^i$ through the relation $(r_{j,m}^i, h_m)$. In the case of Figure 2, *Crag Sandburg High Shool*, *Evan Jager* are the connected entities of *Crag Sandburg High Shool* after entity discovery. A **context-based entity prune** step will be executed later to select top-$W$ entities from all the connected entities as new topic entities $\mathcal{E}_{topic}^{i+1}$, thereby completing the entire graph retrieval step.

### 3.2.2 KNOWLEDGE-GUIDED CONTEXT RETRIEVAL

In this step, ToG-2 digs granular information following high-level knowledge guidance provided by the KG. Once we get all candidate entities by Function 4, we collect the documents relevant to each candidate entity $c_{j,m}^i$, forming a context pool of candidate entities for the current iteration. In the case of Figure 2, the context pool of the first iteration contains documents relevant to candidate entities including *Crag Sandburg High School*, *Evan Jager*, *Lebanon High School*, *1984 Summer Olympics* and so on.

**Entity-guided Context Retrieval:** In order to find useful information within the document context of candidate entities, we apply DRMs to calculate the relevance scores of paragraphs. Rather than directly calculating relevance scores between a context and the question—thereby neglecting the relationship between each context and its corresponding entity—we translate the current triple $Pc_{j,m}^i = (e_j^i, r_{j,m}^i)$ of candidate entity $c_{j,m}^i$ into a brief sentence and append it to the context pending score calculation. Formally, we formulate the relevance score of $z$-th chunk of $c_{j,m}^i$ as:

$$s_{j,m,z}^i = \text{DRM}(q, [triple\_sentence(Pc_{j,m}^i) : chunk_{j,m,z}^i]). \tag{5}$$

Then the top-$K$ scored chunks $Ctx^i$ will be chosen as references for the reasoning phase.

**Context-based Entity Prune:** The selection of candidate entities is based on the rank scores of their context chunks. The ranking score of a candidate entity $c_{j,m}^i$ is calculated as the exponentially decayed weighted sum of the scores of its chunks that rank in top-$K$ (K denoted the Context Number for reasoning), which is formally formulated as:

$$\text{score}(c_{j,m}^i) = \sum_{k=1}^{K} s_k \cdot w_k \cdot \mathbb{I} \text{ (the $k$-th ranked chunk is from $c_{j,m}^i$)}, \tag{6}$$

where $w_k = e^{-\alpha \cdot k}$, $s_k$ is the score of the $k$-th ranked chunk, $\mathbb{I}$ is the indicator function that equals 1 if the $k$-th chunk belongs to $e_{j,m}^i$, and $K$ and $\alpha$ are hyperparameters. Candidate entities with top-$W$ scores will be selected as the topic entities $\mathcal{E}_{topic}^{i+1}$ in the next iteration. Candidate Entities with low relevance scores like *Evan Jager* in the Figure 2 are pruned.

### 3.3 Reasoning with Hybrid Knowledge

At the end of the $i$-th iteration, we prompt LLM with all knowledge found, including $Clues^{i-1}$, triple paths, top-$K$ entities and the corresponding context chunks, to evaluate if the given knowledge is sufficient to answer the question, where $Clues^{i-1}$ is the retrieval feedback from the previous iteration, aiming to maintain useful knowledge in historical context. If LLM judges the provided knowledge is enough to answer the question, it directly outputs the answer. Otherwise, we prompt LLM to output helpful clues $Clues^i$ summarized from existing knowledge and then reconstruct an optimized query based on accurate information until the maximum depth $D$ is reached. The process is formulated as:

$$\text{PROMPT}_{rs}(q, \mathcal{P}^i, Ctx^i, Clues^{i-1}) = \begin{cases} Ans., & \text{if the knowledge is sufficient.} \\ Clues^i, & \text{otherwise.} \end{cases} \tag{7}$$

The details of the prompts are shown in Appendix E Table 18 and 19.

## 4 Experiments

### 4.1 Datasets and Metrics

We evaluated our method on different knowledge-intensive reasoning benchmark datasets, including two multi-hop KBQA datasets WebQSP (Yih et al., 2016) and QALD10-en (Usbeck et al., 2023), a multi-hop complex document QA dataset AdvHotpotQA (Ye & Durrett, 2022) which is a challenging subset of HotpotQA (Yang et al., 2018), a slot filling dataset Zero-Shot RE(Petroni et al., 2021), and two fact verification dataset FEVER (Thorne et al., 2018) and Creak (Onoe et al., 2021). The evaluation metric for FEVER and Creak is Accuracy (Acc.), while the metric for other datasets is Exact Match (EM). Recall and F1 scores are not used since knowledge sources are not limited to document databases. Following previous work (Li et al., 2024c), full Wikipedia and Wikidata are used as unstructured and structured knowledge sources for all of these datasets. Compared to the distractor setting, this full Wiki setting makes the retrieval process more challenging and can better evaluate the effectiveness of various methods on knowledge reasoning tasks.

Since Wikipedia is commonly used for training LLMs, a domain-specific QA dataset that has not been exposed during the pre-training processes of the LLMs is needed for better evaluating different LLM-based approaches. We collect thousands of 2023-year Chinese financial statements as a new corpus, and further build a KG and manually design 97 multi-hop questions based on the corpus. In this paper, we refer to this dataset as ToG-FinQA. Companies and other organizations are extracted as entities in the KG, with 7 types of relationships defined (*subsidiary company*, *main business*, *supplier*, *sibling company*, *bulk transaction*, *subsidiary*, and *customer*). Financial statements are used as entity contexts. Please refer to Appendix C for detailed information of ToG-FinQA.

### 4.2 Baselines

We compare ToG-2 with both widely-used baselines and state-of-the-art methods to provide a more comprehensive overview: 1) LLM-only methods without external knowledge, including Direct Rea-

| Baseline Type | Method | Datasets | | | | | |
|---|---|---|---|---|---|---|---|
| | | WebQSP (EM) | AdvHotpotQA (EM) | QALD-10-en (EM) | FEVER (Acc.) | Creak (Acc.) | Zero-Shot RE (EM) |
| LLM-only | Direct | 65.9% | 23.1% | 42.0% | 51.8% | 89.7% | 27.7% |
| | CoT | 59.9% | 30.8% | 42.9% | 57.8% | 90.1% | 28.8% |
| | CoT-SC | 61.1% | 34.4% | 45.3% | 59.9%[†] (56.2%[‡]) | 90.8% | 45.4% |
| Text-based RAG | Vanilla RAG | 67.9% | 23.7% | 42.4% | 53.8% | 89.7% | 29.5% |
| KG-based RAG | ToG | 76.2% | 26.3% | 50.2% | 52.7% | **93.8%** | 88.0% |
| Hybrid RAG | CoK | 77.6% | 35.4%[‡] (34.1%[†]) | 47.1% | **63.5%**[†] (58.5%[‡]) | 90.4% | 75.5% |
| Proposed | ToG-2 | **81.1%** | **42.9%** | **54.1%** | 63.1[†] (**59.7%**[‡]) | 93.5% | **91.0%** |

Table 1: Performance comparison of different methods with GPT-3.5-turbo across various datasets. Note that the CoK model has 6-shot and 3-shot settings. We present the best performance of CoK under different shot settings for each dataset. For AdvHotpotQA and FEVER, we use the results reported in the original paper of CoK, where † represents the 3-shot setting and ‡ represents the 6-shot setting. Bold numbers represent the highest result under parallel settings.

| Model | ToG-2 | Vanilla RAG | CoT | ToG | GraphRAG |
|---|---|---|---|---|---|
| ToG-FinQA | **34.0%** | 0 | 0 | 14.0% | 6.2% |

Table 2: ToG-FinQA Results

soning, Chain-of-Thought (Wei et al., 2022b), Self-Consistency (Wang et al., 2023a); 2) Vanilla RAG, indicating the text-based RAG method that directly retrieves from entity documents and answers the question; 3) KG-based RAG method: Think-on-Graph (Sun et al., 2024), a KG-based RAG method that searches useful KG triples for reasoning; 4) Chain-of-Knowledge (Li et al., 2024c), a hybrid RAG method retrieving knowledge from Wikipedia, Wikidata and Wikitable; 5) GraphRAG (Edge et al., 2024), a hybrid RAG method that first builds a KG from documents and enables KG-enhanced text retrieval. Notably, we only evaluate GraphRAG on ToG-FinQA due to its unaffordable computation cost during the indexing process on a large corpus. For a fair comparison, all baselines are used with GPT-3.5-turbo and evaluated under an unsupervised setting.

## 4.3 IMPLEMENTATION DETAILS

We mainly use GPT-3.5-turbo as the backbone LLM for a fair comparison to other baselines. To evaluate the effect of backbone LLMs on ToG-2's performance, we conduct experiments with GPT-4o, Llama3-8B and Qwen2-7B. Consistent with the ToG settings, we set the temperature parameter to 0 for all generations. The width $W = 3$ and the maximum number of iterations (depth) is 3. During relation prune, we set a threshold of 0.2 to filter relations with low relevance scores and then select the top $W$. For context retrieval, we utilize the BGE-embedding model without any fine-tuning. In entity prune, we maintain 10 ($K = 10$) top-scored sentences for calculating entity scores. In the reasoning stage, we employ a 2-shot demonstration for most of the datasets. Following CoK (Li et al., 2024c), we employ 3-shot and 6-shot demonstration for FEVER and make the LLM execute a self-consistency reasoning step first.

## 4.4 MAIN RESULTS

The main results of several open-source datasets are shown in Table 1. We note that ToG-2 outperforms other baselines on WebQSP, AdvHotpotQA, QALD-10-en, and Zero-Shot RE. Notably, WebQSP, AdvHotpotQA and QALD-10-en are multi-hop reasoning datasets. On Fever, ToG-2 has a competitive performance to CoK since the fact statements in Fever mainly are about single-hop relations and thus do not need in-depth information retrieval. In another fact verification dataset Creak, all fact statements can be verified based on Wikidata. Thus, ToG-2 and ToG have similar performances on Creak. Meanwhile, compared to original ToG, ToG-2 achieved a substantial improvement on AdvHotpotQA (16.6%) and also demonstrated notable enhancements on other datasets (4.93% on WebQSP, 3.85% on QALD-10-en, 3% on Zero-Shot RE, and 10.4% on FEVER). This demonstrates the advantages of our KG×Text RAG framework in addressing complex problems.

| | Llama-3-8B | | Qwen2-7B | | GPT-3.5-turbo | | GPT-4o | |
|---|---|---|---|---|---|---|---|---|
| | Direct | ToG-2 | Direct | ToG-2 | Direct | ToG-2 | Direct | ToG-2 |
| AdvHotpotQA | 20.8 | 34.7 (**66.8%** ↑) | 17.9 | 30.8 (**72.1%** ↑) | 23.1 | 42.9 (**85.7%** ↑) | 47.7 | 53.3 (**11.3%** ↑) |
| FEVER | 35.5 | 52.9 (**49.0%** ↑) | 38.6 | 53.1 (**38.1%** ↑) | 51.8 | 63.1 (**21.8%** ↑) | 66.2 | 70.1 (**5.9%** ↑) |
| ToG-FinQA | 0 | 8.2 | 0 | 10.3 | 0 | 34.0 | 0 | 36.1 |

Table 3: Performance comparison of direct reasoning and ToG-2 with different backbone models.

Most LLMs have been pre-trained on Wikipedia which is also the unstructured knowledge source of datasets like AdvHotpotQA and Fever, and thus LLM-only methods can work on these datasets. To further evaluate the effectiveness of different methods on domain-specific reasoning scenarios where it is essential to retrieve information from specific knowledge sources since LLMs themselves usually lack relevant knowledge, we conduct an extra evaluation in our established ToG-FinQA dataset. Table 2 shows a comparison of ToG-2 with CoT and the original RAG based on GPT-3.5-turbo. ToG-2 demonstrates a notable advantage over other baselines in this task. Both vanilla RAG and CoT struggled, indicating that traditional RAG and CoT methods may not enable LLMs to solve unseen complex problems. GraphRAG only answered correctly 6.2%, suggesting that while loosely coupled hybrid RAG can retrieve information from both knowledge graphs and documents, it fails to perform multi-hop context retrieval and reasoning with the help of the KG. Compared to GraphRAG, ToG shows improved effectiveness, albeit limited, indicating that the knowledge framework provided by triples can better support multi-hop reasoning.

## 4.5 ABLATION STUDY

### 4.5.1 ABLATION OF LLM BACKBONES

**To what extent do LLMs with varying capabilities benefit from the knowledge enhancement of ToG-2?** To answer this research question, we analyzed the enhancement of ToG-2 under different LLM backbone selections through experiments on AdvHotpotQA and FEVER. The experimental results are shown in Table 3.

We observe that ToG-2 can elevate the reasoning capability of weaker LLMs, e.g., Llama-3-8B, Qwen2-7B to the level of direct reasoning by more powerful LLMs, e.g., GPT-3.5-turbo, which supports our intuition that ToG-2 helps LLMs with knowledge and comprehension bottlenecks. On the other hand, powerful LLMs, e.g., GPT-3.5-turbo and GPT-4o, can still benefit from ToG-2 to improve their own performances on complex knowledge reasoning tasks, indicating that ToG-2 may achieve even higher performance with stronger LLMs. On AdvHotpotQA and FEVER which use Wikipedia as knowledge sources, the most powerful LLMs among all backbone LLMs, GPT-4o experiences the least improvement since GPT-4o has a better memory for knowledge related to Wikipedia than other backbone LLMs, making its direct reasoning performance on these datasets already desirable. It is evident that LLMs with stronger reasoning ability like GPT-4o benefit more significantly from ToG-2 on ToG-FinQA when the relevant knowledge has not been exposed during pre-training and a knowledge retrieval process is necessary. In such scenarios, ToG-2 enables more in-depth knowledge retrieval and more reliable reasoning with stronger LLMs.

### 4.5.2 CHOICES OF ENTITY-PRUNING TOOLS

At the step of **context-based entity prune**, the selection of new topic entities should be based on the relevance of the corresponding contexts, which then reversely guides the next round of graph path exploration. In this experiment, we evaluate different methods for relevance scoring on a sampled set from AdvHotpotQA. In addition to the three DRM models—BGE-Embedding, BGE-reranker, and Minilm—we also tested the sparse retrieval model BM25 and generative ranking with LLMs.

**What kind of tools are most suitable for tight coupling graph-based knowledge and document-based knowledge?** First, Figure 3a presents various choices of pruning tools with a fixed number $K = 10$ of entity contexts. The BGE-Reranker demonstrates the best performance, followed by Minilm, which serves as a lighter alternative to the former. Generative-based and embedding-based tools show comparable results, while BM25, as a classic and efficient retrieval method, exhibits performance close to that of the advanced models. Considering the long runtime of LLMs and the lower accuracy of embedding models, employing a reranker for entity pruning strikes the best

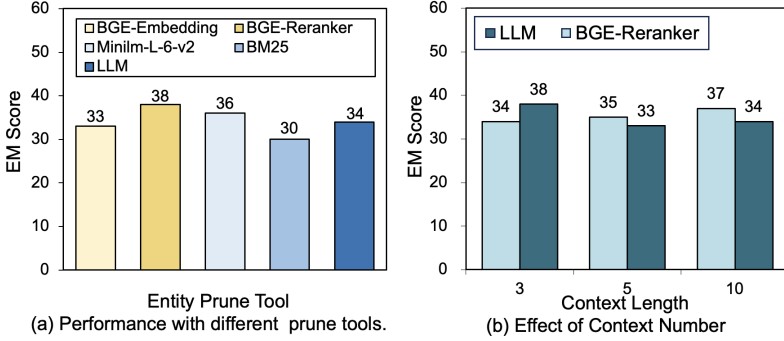

Figure 3: Comparative analysis of Entity-Pruning Tools.

balance of both. The observation also suggests that classic retrieval techniques like BM25 alongside more sophisticated models can still yield good results in a resource-efficient manner.

Then we conduct another experiment in Figure 3b to find the best setting of context numbers $K$. Our findings indicate that BGE-Reranker tends to yield better results with longer context inputs, whereas LLM shows diminished effectiveness. Nevertheless, the overall performance of the pruning tool using BGE-Reranker slightly exceeds that of the LLM. We attribute this outcome to the fact that reasonably more contexts allow for greater tolerance toward irrelevant documents, and DRMs can handle a large number of candidates. While LLM possesses strong discriminative capabilities and offers greater flexibility, its performance is significantly constrained by the input length. Thus, we opt for entity pruning based on the DRM approach, considering its advantages in cost-effectiveness and enhanced generalization.

### 4.5.3 WIDTH & DEPTH SETTINGS

**Is a broader exploration scope necessarily better?** To answer this research question, we analyze the impacts of maximum width $W$ and depth $D$ settings. Figure 4 illustrates the performance of ToG-2 on AdvHotpotQA across varying maximum inference width and depth configurations. The results demonstrate a gradual improvement in model performance as the width increases from 2; however, the marginal gains diminish beyond a width of 3. This trend is at-

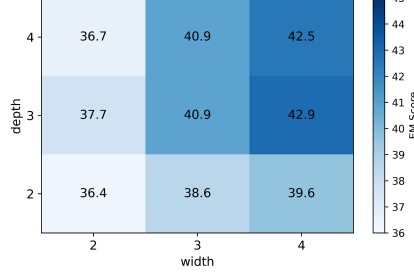

Figure 4: Width and depth settings analysis.

tributed to the inference width, which corresponds to the number of top-$W$ relevant entities retained at each step; a greater width allows for more leniency in topic pruning. With respect to depth, the model's performance plateaus beyond a depth of 3. These observations suggest that a larger search scope is not always better; it should be adjusted according to the difficulty of the task.

### 4.6 RUNTIME ANALYSIS

In the above-mentioned experiments, we use Equation 3 in Relation Prune for efficiency. Under this setting, the reasoning process of ToG-2 needs at most $2D+(D-1)+1$ calls to the LLM. When using Equation 2 in relation pruning, ToG-2 needs at most $WD + D + 1$ LLM calls and can achieve slightly higher performances on several datasets as shown in Appendix B.3 Table 7. In Table 7, we also report the results of an additional ablation study. By contrast, ToG needs at most $2WD + D + 1$ LLM calls. We compare the specific runtime between ToG and ToG-2 on different datasets in Appendix B.4 Table 8. Even though ToG-2 integrates additional knowledge through entity context, ToG-2 still manages to

| Answer Type | Percentage |
|---|---|
| Direct Answer | 16.13% |
| Triple-enhanced Answer | 9.68% |
| Doc-enhanced Answer | 41.94% |
| Both-enhanced Answer | 32.26% |

Table 4: Sources of answer clue origins: **Direct Answer** refers to questions that can be answered directly without additional information. **Triple-enhanced Answer** refers to answers that use triple links as clues. **Doc-enhanced Answer** refers to answers where entity context information provides clues for generating the answer. **Both-enhanced Answer** refers to answers where both triple links and entity context information contribute.

reduce the entity pruning phase to 68.7% of ToG's runtime on average. This is because ToG relies on LLM-based entity pruning, while ToG-2 leverages DRMs, which can handle more candidates and compute relevance scores faster.

## 4.7 MANUAL ANALYSIS

**To what extent does ToG-2 leverage the triple links and the contextual information?** To gain a deeper understanding of ToG-2's behavior, we randomly select 50 AdvHotpotQA reasoning results from ToG-2 and perform a manual analysis about how triple link reasoning and the contextual information of entities within the links contribute to the correct answers, respectively. As shown in Table 4, Doc-enhanced Answers have the highest proportion, at approximately 42%, while Triple-enhanced Answers are the least common, indicating that textual context is often the most important source of information for complex QA tasks. Triple links alone lack detailed context, making it difficult to provide deeper insights, and their role is more of a macro-level guide. Even without relying on the text information within the triples, key clues can still be navigated effectively. The proportion of Both-enhanced Answer shows significant utilization, suggesting that the combination of triple-link reasoning and entity context documents is a highly effective pattern. Direct Answers account for 16% of the responses, showing that for complex questions, the LLM can directly answer relatively few questions and still relies heavily on advanced information-enhanced pipelines.

We demonstrate two cases in AdvHotpotQA to show examples of Both-enhanced Answer and Doc-enhanced Answer. In the Both-enhanced case, the method not only derives that *"the group of Black Indians associated with the Seminole people is Black Seminoles"* from the triple links information, but also leverages the context within the document of *Black Seminoles* to determine their settlement location. In the Doc-enhanced case, the approach utilizes triple links as guidance, ultimately identifying the target information from the document context of *Billy Corgan*, a member of the band in the album. More cases in AdvHotpotQA and ToG-FinQA are shown in Appendix D.

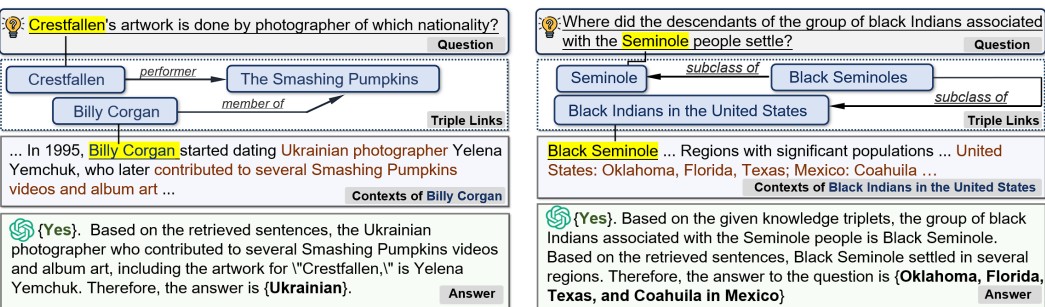

Figure 5: Illustrations of Both-enhanced Answer (left side) and Doc-enhanced Answer (right side).

We also compare the outputs of ToG-2 and CoT regarding the selected questions. As shown in Appendix B.1 Table 5, ToG-2 alleviates the hallucination issues of LLMs compared to CoT with in-depth knowledge retrieval, and tends to refuse answers more often when faced with insufficient knowledge. We also observe a significant number of false negatives in the responses of ToG-2 when using EM as an evaluation metric, suggesting there is still untapped potential for our method.

## 5 CONCLUSION

Existing RAG systems based on KGs or texts struggle to ensure in-depth knowledge retrieval when dealing with complex knowledge reasoning tasks. In this work, we introduce a hybrid RAG paradigm, KG×Text RAG, which tightly couples KG-based and text-based RAG, and propose the Think-on-Graph 2.0 (ToG-2) algorithmic framework that enables reliable graph retrieval by using textual contexts, achieves knowledge-guided context retrieval via the KG, and iteratively perform a cooperative retrieval process to obtain in-depth knowledge for achieving deep and faithful LLM reasoning. Experimental results demonstrate that ToG-2 can significantly improve LLMs of different sizes and outperform existing various LLM reasoning methods and RAG methods without additional training costs.

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

# A PSEUDOCODE

---

**Algorithm 1:** The ToG-2 Algorithm

---

**Input:** Question $q$
**Output:** Answer $Ans$

`// `**`Initialization`**`;`
$Clues = $ **Null**;
Topic entities $\mathcal{E}^0_{topic} \leftarrow$ Perform NER then Topic Prune;
$Ctx^0 \leftarrow$ Retrieve top-$k$ chunks from documents associated with $\mathcal{E}^0_{topic}$ using DRM;
Prompt LLM evaluates the sufficiency of knowledge to answer $q$:
  PROMPT$_{rs}(q, Ctx^0, Clues^0)$
**if** *Information is sufficient* **then**
  | Output $Ans$;
  | **return**
**end**
**else**
  | Outputs $Clues^0$;
**end**
**for** $i = 1$ *to* $D$ **do**
  | `// `**`Context Enhanced Graph Retrieval`**
  | **for** $e^i_j \in \mathcal{E}^i_{topic}$ **do**
  |   | Find relations $Edge(e^i_j) = \{(r^i_{j,m}, h_m)\}$;
  | **end**
  | Use LLM to select and score relations: $\mathcal{R}^i \leftarrow$ PROMPT$_{RP\_cmb}(\mathcal{E}^i_{topic}, q, \{Edge(e^i_j)\}^W_{j=1})$;
  | **for** $r^i_{j,m} \in \mathcal{R}^i$ **do**
  |   | Find connected entities $c^i_{j,m} = \text{Tail}(e^i_j, (r^i_{j,m}, h_m))$;
  |   | Collect documents of candidate entities $c^i_{j,m}$ and split into chunks;
  | **end**
  |
  | `// `**`Knowledge-guided Context Retrieval`**
  | **for** $chunk^i_{j,m,z}$ *of* $c^i_{j,m}$ **do**
  |   | Compute relevance score $s^i_{j,m,z} = \text{DRM}(q, [triple\_sentence(Pc^i_{j,m}) : chunk^i_{j,m,z}])$;
  | **end**
  | $Ctx^i \leftarrow$ Select top-$K$ scored chunks;
  | Compute scores for candidate entities based on chunk scores: score$(c^i_{j,m})$;
  | $\mathcal{E}^{i+1}_{topic} \leftarrow$ Select top-$W$ entities;
  |
  | `// `**`Reasoning with Hybrid Knowledge`**
  | Prompt LLM evaluates sufficiency of knowledge to answer $q$:
  | PROMPT$_{rs}(q, \mathcal{P}^i, Ctx^i, Clues^{i-1})$
  | **if** *Knowledge is sufficient* **then**
  |   | Output $Ans$;
  |   | **return**
  | **end**
  | **else**
  |   | Outputs $Clues^i$;
  | **end**
**end**
Output failure or best effort answer;

---

| Category | Type | Definition | ToG-2 w/o SC | GPT3.5-SC |
|---|---|---|---|---|
| **Correct** | Correct | The answer is correct. | 100% | 88.3% |
| | False positive | Incorrectly judged as correct. | 0% | 11.7% |
| **Incorrect** | False negative | Correct answer but judged as incorrect due to ambiguous questions or labels. | 38.7% | 18.1% |
| | Hallucination | Incorrect answer due to hallucination. | 9.7% | 72.7% |
| | No Info. | Relevant information missing or not found in knowledge sources. | 48.3% | — |
| | Misunderstood | LLM misunderstood the question and gave an irrelevant response. | 3.2% | 9.1% |

Table 5: Human study for Correct and Incorrect cases from randomly sampled HotpotQA questions.

# B  ADDITIONAL EXPERIMENT ANALYSIS

## B.1  MANUAL ANALYSIS (CONTINUE)

The manual analysis results presented in Table 5 reveal several additional observations:

**1)** Even though ToG-2 consistently outperforms the baselines across various datasets, it still faces significant bottlenecks when it comes to information retrieval. These issues are largely due to incomplete knowledge sources and the limitations of retrieval models. For instance, Wikidata often contains inaccuracies, contradictions, or gaps, and many Wiki entities are missing key details on their Wikipedia pages. On the retrieval side, different types of questions require different information, and a one-size-fits-all retrieval model can struggle without specific training. Therefore, ToG-2 can be further optimized through plug-ins that incorporate more comprehensive knowledge graphs and more advanced retrieval models, which is convenient.

**2)** When handling less straightforward or more complex questions, LLMs can sometimes misunderstand the intent. For example, in the question "*What number president nominated Annie Caputo to become a member of the Nuclear Regulatory Commission?*" the correct answer is *45th* but the model might answer *Donald Trump* (which, while true, isn't exactly what the question was asking).

**3)** EM (Exact Match), while often used to assess model performance on QA problems, is not ideal to handle the complexity of such generating. EM struggles to match aliases correctly, and also can't handle answers with different levels of detail. For example, if the question is "*Where was A born?*" without specifying the level of detail, both *New York* and *Manhattan* should be accepted as correct answers. Researchers have attempted to use LLMs as evaluators, considering factors such as efficiency, cost, and the potential bias in LLM evaluations, we opted to use EM for a rough evaluation. While the role of the evaluator is indeed a significant challenge in the era of large language models, this work does not delve into that aspect.

## B.2  SUPPLEMENTARY ANALYSIS FOR THE MAIN RESULTS

As we observed a performance discrepancy between ToG and ToG-2 on the Creak dataset in Table 1, we aim to provide a more detailed analysis on this particular dataset. This analysis will offer insights into the underlying factors contributing to the observed discrepancy. Furthermore, we will showcase and discuss the implications of these findings for understanding the strengths and limitations of both methods in the context of the Creak dataset. Specifically, we manually analyzed the reasoning process for 50 examples. Table 6 provides an overview of the distribution of answer types for ToG-2, following the format of Table 4 in our paper.

| Answer Type | ToG-2 |
|---|---|
| Triple-enhanced Answer | 42% |
| Doc-enhanced Answer | 24% |
| Both-enhanced Answer | 34% |

Table 6: Distribution of answer types for ToG-2 on the Creak dataset.

Due to the relatively simple nature of the questions in the Creak dataset, results can often be inferred directly using triples. As a result, ToG already performs well on this dataset, and the richer entity document information utilized by ToG-2 does not provide significant additional benefits. As shown in the table, ToG-2 produces a higher proportion of Triple-enhanced Answers on Creak compared to more complex datasets like HotpotQA.

We conducted a detailed case study of the behavior of ToG and ToG-2, highlighting three representative cases that elucidate the primary reasons for the observed performance differences. These bad cases reveal distinct patterns in how each model approaches reasoning.

---

**Case 1:** *One can watch Bugs Bunny on the television.*

**ToG-2 Response:** {Yes}, Bugs Bunny can be watched on television. The references mention the history and success of the Bugs Bunny series, indicating its presence on various channels and streaming services over the years. The knowledge triplets also confirm Bugs Bunny's association with iconic works like Looney Tunes and notable creators like Tex Avery and Mel Blanc.

**ToG Response:** {Yes}, the claim is {True}. Based on the knowledge triplets, Bugs Bunny is a beloved character from iconic works such as Looney Tunes, which was created by Warner Bros, and has been featured extensively in television broadcasts since its creation.

**Label:** True

**Error Reason:** The model failed to produce the expected output format {True}. The LLM did not strictly follow the instructions, which is a low-probability event but accounts for a large proportion of errors.

---

**Case 2:** *Quark served the man a drink on Star Trek: Deep Space Nine.*

**ToG-2 Response:** {No}, the claim is {False}. The claim that "Quark served the man a drink on Star Trek: Deep Space Nine" cannot be evaluated based on the given information. The references provide information about the character Quark is a Ferengi bartender on Deep Space Nine, but there is no specific mention of him serving a drink to a man.

**ToG Response:** {Yes}, the claim is {True}. The knowledge triplets confirm Quark's association with Star Trek: Deep Space Nine and his role as a bartender, serving drinks to patrons in the series.

**Label:** True

**Error Reason:** The model's failure stems from two main issues: overcautiousness and semantic ambiguity. Despite identifying the golden sentence, "Quark was the bartender on Deep Space Nine," ToG-2's excessive caution prevented it from making implicit inferences without explicit evidence. Additionally, the phrase "the man" implies a specific individual without sufficient context, leading to a misinterpretation.

---

**Case 3:** *Parkinson's disease causes various hallucinations in people.*

**ToG-2 Response:** {Yes}, the claim is {True}. The retrieved references mention that Parkinson's disease psychosis (PDP) is present in around 20 percent of cases and comprises hallucinations, illusions, and delusions. This confirms that Parkinson's disease can indeed cause various hallucinations in people.

**ToG Response:** {No}, the claim is {False}. The retrieved references do not provide any evidence linking Parkinson's disease to causing various hallucinations in people. As no supporting information was found, it is concluded that this condition does not lead to hallucinations.

**Label:** False

**Error Reason:** Although ToG-2's response appears reasonable from a medical perspective, the golden label is "False." This misalignment suggests that ToG-2's detailed document analysis can sometimes lead to conclusions that diverge from the dataset's annotations, particularly when the dataset lacks nuanced inferences.

---

**Conclusion:** ToG performs well in scenarios that demand rapid, triplet-based reasoning, while ToG-2 demonstrates its strength in cases requiring detailed contextual reasoning. However, ToG-2's overcautious behavior and occasional divergence from the ground truth reveal its tendency to prevent hallucinations at the cost of misalignment with less explicit labels. Additionally, the quality issues within some questions in the Creak dataset also limit the upper bound of model performance.

### B.3 SUPPLEMENTARY ABLATION STUDIES

To evaluate the contribution of each component in ToG-2, we conducted comprehensive ablation experiments across WebQSP, AdvHotpotQA, QALD-10-en and FEVER. To conserve API costs, we sampled a small subset of the FEVER dataset for ablation experiments. Compared to the performance on the other three datasets, on WebQSP, the effectiveness of Topic Prune (TP) is more pronounced, possibly due to the higher relative proportion of general entities in WebQSP questions, which tends to introduce more unnecessary noise. Additionally, the retrieval feedback (Clue) also brought relatively consistent improvements across each dataset, indicating that adaptive query optimization can help the LLM better understand the tasks.

| Model | Datasets | | | |
|---|---|---|---|---|
| | WebQSP (EM) | AdvHotpotQA (EM) | QALD-10-en (EM) | FEVER (Acc.) |
| Direct | 65.9% | 23.1% | 42.0% | 52.1% |
| ToG-2$_{/TP/RC/Clue}$ | 78.7% | 40.9% | 51.1% | 56.3% |
| ToG-2$_{/TP/Clue}$ | 78.0% | 40.2% | 49.9% | 56.0% |
| ToG-2$_{/TP}$ | 77.6% | 41.9% | 52.9% | 56.5% |
| ToG-2 | **81.1%** | **42.8%** | **54.1%** | **58.5%** |

Table 7: Influence of each module in ToG-2 on the final performance. / denote without. $Clue$: the retrieval feedback. $TP$: topic prune. $/RC$: replace relation prune Eq.3 with Eq.2.

### B.4 RUNTIME ANALYSIS RESULTS

**Does ToG-2 improve graph exploring efficiency compared to ToG?** Therefore, we first statistically compared the runtime of ToG and ToG-2 across different stages under the same settings. Specifically, we analyze the average number of iterations in answering each discrete question, along with the average runtime for each iteration.

The results in Table 8 show a substantial improvement in ToG-2's relation pruning time, reducing it to just 45% of ToG's. This gain comes from the relation pruning combination strategy, which cuts down the number of LLM calls from W times to 1 time each iteration. Even though ToG-2 integrates additional knowledge through entity context, ToG-2 still manages to reduce the entity pruning phase to 68.7% of ToG's runtime on average across both datasets. This is because ToG relies on LLM-based entity pruning, while ToG-2 leverages DRMs, which can handle more candidates and compute relevance scores faster.

To better illustrate the efficiency of our method compared to the baselines, we analyze the average total runtime and API call counts for each method on HotpotQA. The results are summarized in Table 9 below. ToG: For each question, ToG requires up to $2WD + D + 1$ API calls, where $W$ is the

| Dataset | Method | Avg. time per RP (s) | Avg. time per EP (s) | Avg. time per Reasoning (s) | Avg. Depth |
|---|---|---|---|---|---|
| FEVER | ToG-2 | 6.4 | 4.5 | 5.9 | 2.21 |
| | ToG | 14.2 | 6.5 | 5.5 | 1.94 |
| AdvHotpotQA | ToG-2 | 6.1 | 4.3 | 5.7 | 2.17 |
| | ToG | 13.8 | 6.3 | 5.4 | 1.71 |

Table 8: Running time comparison of ToG-2 and ToG on FEVER and AdvHotpotQA datasets.

| Method | Avg Total Time (s) | API Calls |
|---|---|---|
| ToG-2 | 27.3 | 5.4 |
| NaiveRAG | 10.2 | 1 |
| ToG | 69.3 | 16.3 |
| CoK | 30.1 | 11 |

Table 9: Comparison of API Call Counts and Runtime Across Methods.

maximum width (number of knowledge triplets explored in parallel) and $D$ is the maximum iteration depth. This structure enables comprehensive multi-hop reasoning and knowledge exploration, but results in a relatively high number of API calls per question.

CoK: To ensure fairness in comparison, we restricted the knowledge domain to factual data from WikiData, Wikipedia, and DPR. We also adopted the `update_rationales_at_once` setting from the original implementation. CoK has a fixed API call count of 11 per question: 1 domain selection call, 1 chain-of-thought sub-question generation call, $2 \times 3$ knowledge retrieval calls, 2 rationale editing calls, and 1 final answer generation call.

NaiveRAG: To simulate a retrieval corpus, we use all entity documents encountered during ToG-2 iterations. NaiveRAG uses a two-stage retrieval system that combines BM25 with BGE-Reranker, resulting in the fewest API calls among the compared methods.

ToG-2: For each question, ToG-2 requires up to $2D + (D-1) + 1$ API calls, where:

- $2D$: Calls for Relation Pruning (RP) and Reasoning during each iteration.
- $D-1$: Query re-writing between iterations.
- 1: Topic Prune (TP).

Compared to baselines, ToG-2 achieves a good balance between cost-efficiency and performance. It reduces the total API calls and runtime while maintaining robust reasoning capabilities.

### B.5 IMPACT OF THRESHOLD SETTING IN RELATION PRUNE

We conducted threshold experiments in Relation Prune (RP) to analyze the effect of different threshold scores. Specifically, we compared the effects of three thresholds (0.2, 0.5, and 0.8) on the final results in the HotpotQA dataset. The results are summarized in Table 10.

| Threshold | Results (%) | Avg. Retained Score |
|---|---|---|
| 0.2 | 42.9 | 0.65 |
| 0.5 | 43.2 | 0.68 |
| 0.8 | 40.0 | 0.92 |

Table 10: Effects of different thresholds on the final results in HotpotQA. The *Avg. retained score* represents the average score of all relations that pass through the threshold filtering.

When the threshold is set between 0.2 and 0.5, its impact on the final result is minimal. However, when the threshold increases to 0.8, the results weaken significantly. We attribute this to two main reasons:

1. During the relation pruning stage, the relations passing the threshold are further filtered by selecting the top relations based on the width $W$. Therefore, even if the threshold is set very low, relations with extremely low scores will ultimately fail to be included in the Top $W$. This explains why the *Avg. retained score* is very close when the threshold is 0.2 or 0.5. However, setting a reasonable threshold is still necessary to prevent the inclusion of entirely irrelevant relations in cases where very few relations are found and are scored low (e.g., as low as 0.1).

2. Relations with moderate relevance scores (e.g., 0.3–0.7) can potentially lead to useful information. A lower threshold allows for more tolerance for such potential. When the threshold is raised to 0.8, it overly amplifies the impact of the LLM's judgment during relation pruning on the entire process, leading to the loss of potentially valuable relations.

### B.6 IMPACT OF SC SETTING

Figure 6 illustrates the impact of varying self-consistency thresholds on the performance of ToG-2 on AdvHotpotQA. As the threshold increases, we observe a corresponding performance improvement. This can be attributed to the stricter consistency demands placed on the LLM's multiple responses as the threshold rises, which more effectively identifies and filters out complex questions that are challenging for the LLM to resolve directly. These difficult cases are then handed over to ToG-2, enabling more accurate responses. Furthermore, while fewer questions are answered during the SC phase, they are answered with a higher degree of certainty.

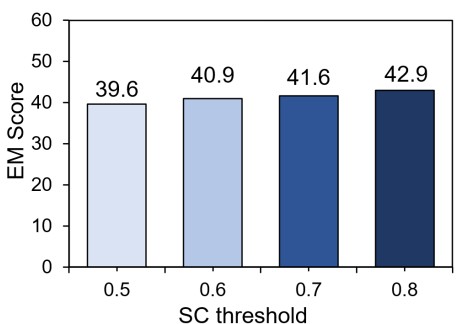

Figure 6: Impact of SC setting

### B.7 IMPACT OF GRAPH COMPLETENESS

To investigate the impact of graph quality on the reasoning performance of ToG-2, we designed an experiment to quantify the specific effect of Knowledge Graph (KG) quality on the model's performance. Specifically, we conducted experiments on a subset of randomly sampled HotpotQA questions (100 in total) using simulated incomplete KGs. The incompleteness was introduced by randomly discarding relations and entities discovered during the Relation Discovery (Equation 1) and Entity Discovery (Equation 4) stages. Specifically, we tested scenarios where 30%, 50%, and 80% of the discovered entities and relations were retained, while the rest were randomly removed. This systematic evaluation provided insights into the model's adaptability to varying levels of KG completeness. The results are shown in Table 11.

| KG Completeness (%) | Exploration Setting | EM (%) |
| --- | --- | --- |
| 100 | Default | 43 |
| 80 | Default | 41 |
| 50 | Default | 35 |
| 30 | Default | 23 |
| 30 | Adjusted | 29 |

Table 11: Impact of KG Completeness on ToG-2.0 Performance with Default ($W = 3$, $D = 3$) and Adjusted ($W = 8$, $D = 2$) Exploration Strategies

At 80% completeness, the model demonstrated robust performance with minimal impact compared to the fully complete KG, indicating its strength at high completeness levels. With 50% completeness, moderate performance degradation was observed, suggesting that ToG-2.0 remains resilient to moderate KG incompleteness. However, at 30% completeness, performance dropped significantly due to sparse connections in the graph, highlighting the challenges posed by severe KG incompleteness.

To mitigate this, adjustments were made to the exploration strategy: By increasing the exploration width ($W$) to 8 and reducing the exploration depth ($D$) to 2, the model was able to broaden its

search scope, compensating for the missing information. This adjustment led to a notable recovery in performance, bringing it closer to an acceptable level despite the significant KG incompleteness. These findings emphasize the model's adaptability and provide insights into potential strategies for handling incomplete KGs.

## B.8 SIGNIFICANCE TEST

Statistical significance is a key factor in evaluating model performance. To assess this, we conducted pairwise t-tests on select datasets where we had access to both ToG-2 and baseline results. However, many of the baseline results reported in Table 1 were directly cited from their respective original papers or other related works. As a result, we do not have access to the detailed outputs of these baseline models for individual questions, preventing us from performing significance tests on those baselines at this time.

For the WebQSP, Zero-Shot-RE, QALD-10-en, and Creak datasets, we computed pairwise t-tests comparing ToG-2 with CoK and ToG as in Table 12.

| (ToG-2 vs. ) Method | WebQSP | Zero-Shot-RE | QALD-10-en | Creak |
|---|---|---|---|---|
| **CoK** | $p < 0.05$ | $p < 0.01$ | $p < 0.05$ | $p < 0.05$ |
| **ToG** | $p < 0.05$ | $p < 0.05$ | $p < 0.05$ | $p > 0.9$ |

Table 12: ToG-2 significantly outperforms CoK and ToG on most datasets ($p < 0.05$), except on Creak where the difference with ToG is not significant ($p > 0.9$), indicating similar performance on this dataset.

## C TOG-FINQA DATASET OVERVIEW

The dataset comprises three main components: (1) a document corpus, (2) a knowledge graph, and (3) QA pairs. Table 13 provides a summary of the key statistics for each component.

| Component | Statistic | Value |
|---|---|---|
| **Document Corpus** | Total documents | 17,013 |
| | Average document length | 40,469 |
| **Knowledge Graph** | Total entities | 671,806 |
| | Total edges | 565,994 |
| | Average edges | 0.84 |
| | Maximum edges | 3,982 |
| | Median edges | 1.0 |
| **QA Pairs** | Total QA pairs | 97 |
| | Topics | Market Analysis, Competitor Analysis, Bulk Transactions, Customer Analysis, Supplier Analysis |

Table 13: Key Statistics of the Chinese Financial QA Dataset

We automatically sample some candidate triple paths based on predefined meta-path rules, such as supplier-customer-supplier (a meta-path that might explore competitive relationships), while preserving the entity context involved along the path. These triple paths, along with their contextual information, are then manually inspected and filtered to select candidates that can logically and factually form multi-hop questions with objective answers.

## D  OTHER CASES

We present two cases of ToG-FinQA in Figure 7. On the left side is a Both-endanced case with the topic of Supplier Analysis. From the triple relationships of predicate relation customers and subject relation suppliers, two suppliers of ZEC were identified, and information related to recycling was found in their documents. On the right side is also a Both-endanced case, candidate entities were identified through paths of the same customer or supplier, and then the context in the documents of the candidate entities was used to confirm that Nanjing Jujong is the target entity.

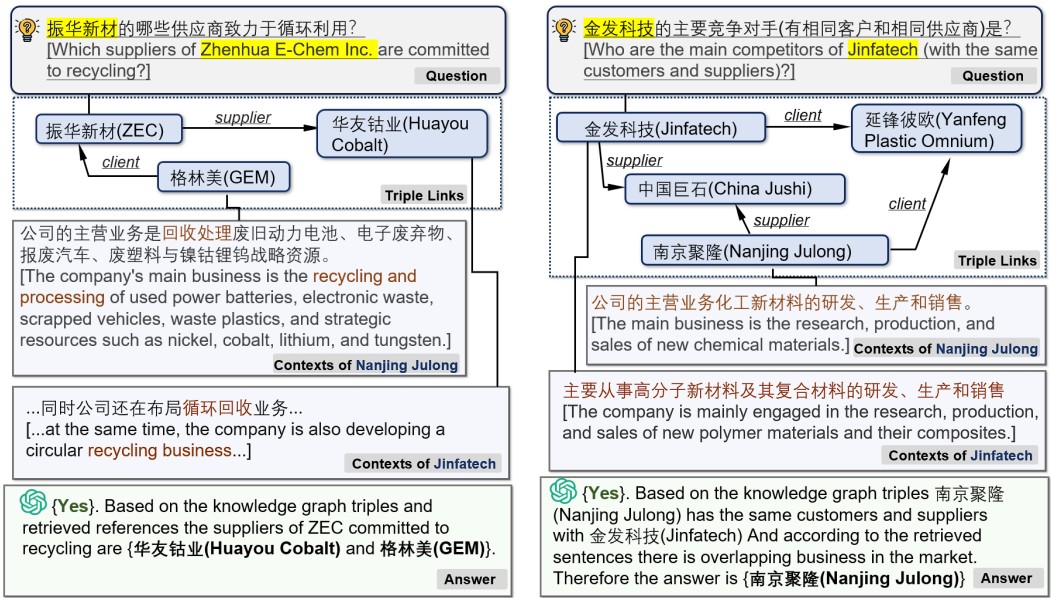

Figure 7: ToG-FinQA Cases

Figure 8 shows a Triple-enhanced case in HotpotQA, where the common style of the two movies is identified by solely utilizing the triple links.

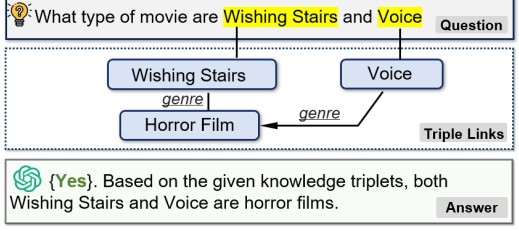

Figure 8: Triple-enhanced Case

# E    PROMPTS

**Topic Prune Prompt**

Given a question and a group of related topic entities derived from the Wikipedia knowledge graph, the task is to select which of these entities are suitable as starting points for reasoning on the Wiki knowledge graph to find information and clues that are useful for answering the question.

Note that your output should be strictly JSON formatted: {id: entity}.

Here are examples showing how to analyze and output a JSON formatted answer:

Example 1:
Question: What major city is the Faith Lutheran Middle School and High School located in?
Topic entities: {
"Q111": "Faith Lutheran Middle School",
"Q39722": "Faith Lutheran High School"
}
Analysis:
All entities are directly related to the core of the question—finding the possible information about Faith Lutheran and its location.
Output:
{"Q111": "Faith Lutheran Middle School", "Q39722": "Faith Lutheran High School"}

Example 2:
Question: How many Turkish verbs ending with "uş" with their lemma?
Topic entities: {
"Q24905": "verb"
}
Analysis:
Q24905 "verb" focuses on verbs, which are action words in a language. The task involves not just identifying verbs but specifically Turkish verbs that end with the suffix "uş,". Therefore, the entity "verb" is too broad and does not point to a specific Turkish verb. Thus there is no suitable topic entity as a starting point for reasoning. Output:
{}

Example 3:
Question: On which island is the Indonesian capital located?
Topic entities: {
"Q252": "Indonesia",
"Q23442": "island"
}
Analysis:
Q252 "Indonesia" represents the country of Indonesia. Considering the question is about the capital of Indonesia and which island it is located on, the entity "Indonesia" is directly related to the core of the question—finding the capital of Indonesia and then identifying the island it is situated on. Therefore, this entity is highly suitable as a starting point for reasoning.
Q23442 "island" represents the concept of an island. While the question does indeed relate to an island, the concept of "island" itself is too broad and does not point to a specific geographical location or country. From the perspective of reasoning on a knowledge graph, trying to find the capital of a specific country based solely on the concept of an island is less relevant and efficient compared to starting from that country. Hence, Q252 "Indonesia" is the more suitable topic entity as a starting point for reasoning on the knowledge graph. It directly connects to the key information of the question and can effectively guide the search for entities related to the answer within the knowledge graph.
Output:
{"Q252": "Indonesia"}

Table 14: Topic Prune

| Relation Prune Prompt |
|---|
| Please retrieve %s relations (separated by semicolon) that contribute to the question and rate their contribution on a scale from 0 to 10. 
 Question: Mesih Pasha's uncle became emperor in what year? 
 Topic Entity: Mesih Pasha 
 Relations: 
 1. child 
 2. country_of_citizenship 
 3. date_of_birth 
 4. family 
 5. father 
 6. languages_spoken 
 7. military_rank 
 8. occupation 
 9. place_of_death 
 10. position_held 

 Answer: 
 1. {family (Score: 10)}: This relation is highly relevant as it can provide information about the family background of Mesih Pasha, including his uncle who became emperor. 
 2. {father (Score: 4)}: Uncle is father's brother, so father might provide some information as well. 
 3. {position_held (Score: 1)}: This relation is moderately relevant as it can provide information about any significant positions held by Mesih Pasha or his uncle that could be related to becoming an emperor. 
 4. ...... 

 Question: Van Andel Institute was founded in part by what American businessman, who was best known as co-founder of the Amway Corporation? 
 Topic Entity: Van Andel Institute 
 Relations: 
 1. affiliation 
 2. country 
 3. donations 
 4. educated_at 
 5. employer 

 Answer: 
 1. affiliation (Score: 8): This relation is relevant because it can provide information about the individuals or organizations associated with the Van Andel Institute, including the American businessman who co-founded the Amway Corporation. 
 2. donations (Score: 3): This relation is relevant because it can provide information about the financial contributions made to the Van Andel Institute, which may include donations from the American businessman in question. 
 3. educated_at (Score: 3): This relation is relevant because it can provide information about the educational background of the American businessman, which may have influenced his involvement in founding the Van Andel Institute. 
 4. ...... |

Table 15: Topic Prune Prompt

---

**Relation Prune CMB Prompt**

---

# Task:
1. Carefully review the question provided.
2. From the list of available relations for their corresponding entity, select the %s that you believe are most likely to link to the entities that can provide the most relevant information to help answer the provided question.
3. For each selected relation, provide a score between 0 to 10 reflecting its usefulness in answering the question, with 10 being most useful.
4. Provide a brief explanation for your choices, highlighting how each selected relation potentially contributes to answering the question.

# The input follows the below format:
Question:[The question text]
Entity 1:[The name of the entity 1]
Available Relations:[A relation list of entity 1 to be chosen.]
Entity 2:[The name of the entity 2]:
Available Relations:[A relation list of entity 2 to be chosen.]
...(Continue in the same manner for additional entities)

# Below is two examples:
# Example1:
Question: Mesih Pasha's uncle became emperor in what year?
Topic Entity: Mesih Pasha
Relations:
1. child
2. country_of_citizenship
3. date_of_birth
4. family
5. father
6. languages_spoken, written_or_signed
7. military_rank
Answer:
1. {family (Score: 1.0)}: This relation is highly relevant as it can provide information about the family background of Mesih Pasha, including his uncle who became emperor.
2. {father (Score: 0.4)}: Uncle is father's brother, so father might provide some information as well.
3. ......

# Example2:
Question: what the attitude of Joe Biden towards China?
Entity 1: china
Relations:
1. alliance
2. international_relation
3. political_system
4. population
Entity 2: joe biden
Relations:
1. political_position
2. presidency
3. family
4. early_life
Answer:
Entity 1:
1. {alliance (Score: 8)}: This relation is highly relevant as it can provide information about the relationship between china and other parties.
2. {political_system (Score: 7)}: This relation is relevant as it can provide information about the policies that might reflect the relationship between china and other parties.
3. ......

---

Table 16: Relation Prune CMB Prompt

---

**Relation Prune CMB Prompt (Continue)**

---

Entity 2:
1. {political_position (Score: 10)}: This relation is highly relevant as it can provide information about Joe Biden's political position to other parties or countries.
2. {presidency (Score: 2)}: This relation is slightly relevant as it can provide information about Joe Biden's position, which might provide clues to answer the question.
3. ......

# For questions involving multiple entities, you are required to analyze the relation list for each entity separately. Then select the %s most relevant relations for each entity, based on the analysis as done in the given example. # It's essential to maintain strict adherence to the line breaks and format as seen in the provided example for clarity and consistency:
Answer:
Entity 1: The Name of Entity 1
1. {Relation1 (Score: X)}: Explanation.
2. {Relation2 (Score: Y)}: Explanation.

Entity 2: The Name of Entity 2
1. {Relation1 (Score: X)}: Explanation
2. {Relation2 (Score: Y)}: Explanation

...(Continue in the same manner for additional entities)

# Now, I will provide you with a new question with %s entities and relations. Please analyze it following all the guidance above carefully.
# For questions involving multiple entities, you are required to analyze the relation list for each entity separately. Then select the %s most relevant relations for each entity, based on the analysis as done in the given example. # It's essential to maintain strict adherence to the line breaks and format as seen in the provided example for clarity and consistency:
Answer:
Entity 1: The Name of Entity 1
1. {Relation1 (Score: X)}: Explanation.
2. {Relation2 (Score: Y)}: Explanation.

Entity 2: The Name of Entity 2
1. {Relation1 (Score: X)}: Explanation
2. {Relation2 (Score: Y)}: Explanation

...(Continue in the same manner for additional entities)

# Now, I will provide you with a new question with %s entities and relations. Please analyze it following all the guidance above carefully.

---

Table 17: Relation Prune CMB Prompt (Continue)

---

**Reasoning Prompt (QA)**

---

Given a question, some clues, the associated retrieved knowledge triplets and related contexts, you are asked to evaluate if these resources, combined with your pre-existing knowledge, are sufficient to formulate an answer ({Yes} or {No}).
Your answer must begin with {Yes} or {No}.
If {Yes}, please note that the analyzed answer entity must be enclosed in curly brackets {xxxxxx}
If {No}, it means the resources are useless or provide clues that are helpful but insufficient to answer the question conclusively. Based on the given knowledge, please summarize any insights (if any) so far that may help answer the question, which must also be enclosed in curly brackets{xxxxxx}.

Here are some examples:
# Example 1:
Question:
The Sentinelese language is the language of people of one of which islands in the Bay of Bengal ?
(Clues):
None
(Knowledge triplets):
Sentinelese language, Indigenous to, Sentinelese people
(Triplet's related context):
The Sentinelese, also known as the Sentinel and the North Sentinel Islanders, are an indigenous people who inhabit North Sentinel Island in the Bay of Bengal in the northeastern Indian Ocean.
(Knowledge triplets):
Bay of Bengal, area, Andaman and Nicobar Islands
(Triplet's related context):
Andaman and Nicobar Islands are an archipelagic island chain in the eastern Indian Ocean across the Bay of Bengal, they are part of India. They are within the union territory of the Andaman and Nicobar Islands
# Answer:
{Yes} The analyzed answer entity is {Andaman and Nicobar Islands}.

# Example 2:
Question:
Who is the coach of the team owned by the most famous English former professional footballer?
(Clues):
The most famous English former professional footballer is David Beckham.
(Knowledge triplets):
David Beckham, co-owned, Inter Miami CF
(Triplet's related context):
Inter Miami CF, founded in 2018, is a professional soccer club based in Miami, Florida. It competes in Major League Soccer (MLS) as part of the Eastern Conference. David Beckham made a 3 years contract with his major coach.
# Answer:
{No} The given knowledge provides helpful information that the footballer is David Beckham and David Beckham's ownership of Inter Miami CF, but they do not explicitly mention who the coach is. The useful information for now is {the team owned by the famous English former professional footballer David Beckham is Inter Miami CF}

Now, please carefully consider the following case:

---

Table 18: Reasoning Prompt (QA).

---

**Reasoning Prompt (QA)**

---

Given a question and some knowledge gained so far, please predict the additional evidence that needs to be found to answer the current question, and then provide a suitable query for retrieving this potential evidence. Note that the query must be included in curly brackets {xxx}.

---

Table 19: Query Re-form Prompt

