# OpenReview forum: "Think-on-Graph 2.0: Deep and Faithful Large Language Model Reasoning with Knowledge-guided Retrieval Augmented Generation"
_ICLR.cc/2025/Conference — ICLR 2025 Poster_

### Official Review · Reviewer_HRGs · 2024-10-28

**Soundness:** 3
**Presentation:** 3
**Contribution:** 3
**Rating:** 6
**Confidence:** 4

**Summary:**

ToG-2 is an iterative RAG framework that combines unstructured and structural knowledge from the input to answer the given question. Concretely, ToG-2 extracts multiple entities from the query, extracts their neighboring entities, filters out these entities via relation pruning to extract a set of candidate entities, and identifies (and prunes) retrieved entity context candidate entity. Experiments show that ToG-2 achieves state-of-the-art on the majority of datasets.

**Strengths:**

State-of-the-art performance is achieved on all but one dataset

ToG-2 conducts extensive experimentation to verify their novelty

**Weaknesses:**

1. Figures

The main figure, as well as the introduction, indicates that ToG-2 is distinctly different from Text-based RAG, KG-based RAG, and Hybrid (text- + KG-based) RAG. Upon inspection, it appears that ToG-2 is just a fancy version of Hybrid RAG in that entities are selected based on a n-order neighborhood of the entities identified in the query. While ToG-2 is no doubt different from previous Hybrid RAG approaches, it should still be considered a Hybrid RAG approach.

The caption of the main figure should give a description of the flow of the methodology. Given that there are multiple modules in ToG-2, and that important information of these modules are not depicted (more in the following comment) an explanation of the information flow is necessary.

While the main figure provides a nice overview of ToG-2, it is missing important detailed information. For example, relation/entity scoring and the prompts used aren’t able to be determined by looking at the main figure. Consider creating another figure that depicts these details.

2. Experiments

No analysis of threshold scores for relation pruning (“The relations with a low score will be pruned” on page 5, lines 230-231). How much would performance vary if the threshold is changed? What does the distribution of relation scores from the LLM look like? These are questions that should’ve been answered. Considering that ToG-2 is a multistage pipeline, and that relation pruning is one of the first steps in the pipeline, analysis here is vital given that error from this module will propagate through your remaining modules. Furthermore, there isn’t any mention (let alone the reasoning) of the threshold score used in ToG-2. In the code, the score that is used is 0.2 (https://anonymous.4open.science/r/ToG2/wiki_func.py, line 38). This should have been mentioned in the manuscript and not have been identified solely by investigating the repository.

Table 6, the influence of each module (which should be in the main content), indicates that the removal of relation pruning increases performance across all datasets (the increase in performance in from row 2 to row 3). This appears to nullify the contribution that relation pruning offers.

Experiments involving LLMs should’ve been repeated multiple times given that LLMs have been shown to vary greatly based on the prompt.

3. Grammatical/formatting errors

On page 1, line 40, there is a space missing between “issues” and “(Zhao et al., 2024)”

The query in the main figure “Where was one of the runners born who almost broke Craig Virgin’s Illinois Boys Cross Country record?” should be re-worded for clarity. Consider stating “What was the birthplace of one of the runners who almost broke Craig Virgin’s Illinois Boys Cross Country record?”.

Consider removing the word “and” on line 228, page 5 between equations 2 and 3, as the current format is an unorthodox way of introducing two equations simultaneously.

“Figure 2” needs a space between “Figure” and “2” on page 5, line 239.

The “a” in “Given a topic entities” in line 241, page 5 should be removed.

The citations in section 4.2 are in double parenthesis. This needs to be fixed.

The title of section 4.5.3 “Width&Depth Settings” is missing a space between “Width” and “&” as well as “&” and “Depth”.

The ablation study comparing the performance when removing different modules (Table 6) should be put in the main content, not in the appendix.

**Questions:**

Did the authors notice that ToG-2 performs better without relation pruning (when topic pruning and retrieval feedback are also removed) than when only topic pruning and retrieval feedback are removed?

It is mentioned that “ToG-2 selectively explores entities neighboring the current topic entities on the KG, using the newly encountered entities to refine the retrieval scope, thereby enhancing both efficiency and accuracy.” (lines 161-180, pages 3-4). How is ToG-2 able to enhance/improve efficiency if there are more entities considered (via neighbors in the knowledge graph) compared to just the entities extracted from the query?

In figure 3b, the performance of ToG-2 when using BGE-Ranker decreases as the number of entities considered increases (from 38% to 33% when the number of entities jumps from 3 to 5). This seems to be counter-intuitive, as the larger the entity space, the more likely it would increase performance (which is the case when using LLMs). Why would this be the case?

Why not consider a dynamic number of candidate topic entities (dynamic based on the iteration)? Logically, it’s more likely that first order neighbors of an entity in the query would be more important than entities in the second order neighborhood (although this assumption might be invalid when dealing with very granular inputs).

What is “CMB” with respect to relation pruning (equation 3, tables 11 & 12)?

---

> ### Author Response · Authors · 2024-11-18
> **Feedback to Official Review by Reviewer HRGs (Part I)**
>
> **We are grateful to the reviewers for their meticulous review and insightful comments, which helped us to refine our paper and ensure its clarity and rigor. Below, we will provide detailed analysis and explanation to address your concerns:**
>
> ### **Response to Weakness 1:**
> We agree with your point that "ToG 2.0 is also a Hybrid RAG." In our paper, we described the existing Hybrid RAG as a "loose-coupling combination" of Text-based RAG and KG-based RAG (as mentioned in line 087), whereas ToG 2.0 represents a "tight-coupling hybrid RAG paradigm" (as mentioned in line 090). The Hybrid RAG described in Figure 1 actually represents the existing loose-coupling hybrid RAG. Due to space limitations in the figure, we omitted "loose-coupling" in the title, but we clarified in the caption of Figure 1 that Figure 1(c) depicts KG+Text Hybrid RAG. We appreciate you pointing out the potential confusion this might cause, and we will update the subheading in future versions to explicitly state that Figure 1(c) refers to KG+Text Hybrid RAG, specifically the loose-coupling hybrid RAG.
>
> We also want to emphasize that Figure 1(d) illustrates an example of KG×Text Hybrid RAG, rather than the complete Think-on-Graph 2.0 framework. Through Figures 1(c) and 1(d), we aim to highlight the distinction between KG+Text (loose-coupling) Hybrid RAG and KG×Text (tight-coupling) Hybrid RAG. The complete Think-on-Graph 2.0 framework is presented in Figure 2. In future revisions, we will refine the subheadings and captions in the figures to make this distinction clearer and avoid any potential misunderstanding for readers.
>
> ### **Response to Weakness 2:**
>
> > No analysis of threshold scores for relation pruning (“The relations with a low score will be pruned” on page 5, lines 230-231). How much would performance vary if the threshold is changed? What does the distribution of relation scores from the LLM look like?
>
> **First,** we conducted threshold experiments in relation pruning. Specifically, we compared the effects of three thresholds (0.2, 0.5, and 0.8) on the final results in HotpotQA. The **Avg. retained score** represents the average score of all relations that pass through the thresholds filtering. The results are as follows:
>
> | Thresholds | Results | Avg. retained score |
> | --- | --- | --- |
> | 0.2 | 42.9% | 0.65 |
> | 0.5 | 43.2% | 0.68 |
> | 0.8 | 40.0% | 0.92 |
>
> When the threshold is set between 0.2 and 0.5, its impact on the final result is minimal. However, when the threshold increases to 0.7, it significantly weakens the results. We attribute this to two main reasons:
>
> 1. During the relation pruning stage, the relations passing the threshold are further filtered by selecting the top relations based on the width WWW. Therefore, even if the threshold is set very low, relations with extremely low scores will ultimately fail to be included in the Top W. This explains why the **Avg. retained score** is very close when the threshold is 0.2 or 0.5. However, setting a reasonable threshold is still necessary to prevent the inclusion of entirely irrelevant relations in cases where very few relations have found and are scored low (e.g., as low as 0.1).
> 2. Relations with moderate relevance scores (e.g., 0.3–0.7) can potentially lead to useful information. A lower threshold allows for more tolerance for such potential. When the threshold is raised to 0.7, it overly amplifies the impact of LLM’s judgement during relation pruning on the entire process, leading to the loss of potentially valuable relations.
>
> By the way, you are absolutely correct that the threshold score used in ToG2.0 (0.2) should have been explicitly mentioned in the manuscript. We will check and ensure that all critical implementation details are clearly documented in the manuscript.
>
> > Table 6, the influence of each module (which should be in the main content), indicates that the removal of relation pruning increases performance across all datasets (the increase in performance in from row 2 to row 3). This appears to nullify the contribution that relation pruning offers.
>
> **Second, regarding the interpretation of** **Table 6**,  we will answer your concern together with **Question 1**.
>
> > Experiments involving LLMs should’ve been repeated multiple times given that LLMs have been shown to vary greatly based on the prompt.
>
> **Finally,** we recognize the potential variability in LLM outputs and agree that repeated experiments can further help quantify it. This design choice is consistent with prior works such as *Think-on-Graph* and *Chain-of-Knowledge*, which also did not conduct multiple runs for LLM-based components.  In our study, we set the temperature to 0 for all generation to ensure reproducibility. Given the deterministic nature of our setup and the associated computational and financial costs of repeated experiments, we believe our approach is reasonable for this context.

---

> ### Author Response · Authors · 2024-11-18
> **Feedback to Official Review by Reviewer HRGs (Part II)**
>
> ### **Response to Weakness 3:**
> Thank you for pointing out the typos and grammatical errors in our paper. We will make the necessary corrections and update the PDF as soon as possible. Previously, due to space constraints, we opted to include Table 6 and the related discussion in the appendix. In future versions, we will follow your suggestion and move this content to the main text.
>
> ### **Response to Questioin 1&5:**
> In Table 6, RC (Relation Prune Combination) refers to relation pruning using Equation 3. In contrast, /RC indicates relation pruning using Equation 2, rather than skipping relation pruning altogether. During the relation pruning process, Equation 2 invokes the LLM API multiple times to rank different relation sets individually, while Equation 3 combines these relation sets and invokes the LLM API only once to rank all relations collectively. As a result, this trade-off makes Equation 3 a more practical choice for real-world applications while maintaining competitive performance.
>
> To ensure the efficiency of our method in practical reasoning scenarios, we adopt Equation 3 for relation pruning in most of our experiments, including the main experiments, ablation studies, and case studies. The second row of Table 6 shows that ToG-2/TP/RC/Clue outperforms ToG-2/TP/Clue, validating our hypothesis that using Equation 2 for relation pruning enhances inference accuracy compared to Equation 3. This point is also noted in the main text (Line 476-478).
>
> > What is “CMB” with respect to relation pruning (equation 3, tables 11 & 12)?
>
> We apologize for the typo in the titles of Tables 11 and 12. The term "Topic Prune CMB Prompt" should have been "Relation Prune CMB Prompt," where "CMB" refers to the relation prune combination (RP_cmb) described in Equation 3. Thank you for pointing this out, and we appreciate your understanding.
>
> ### **Response to Question 2:**
> Since our method involves multiple rounds of context retrieval to address complex reasoning problems, it is crucial to constrain the retrieval scope to improve the efficiency of each retrieval, rather than to directly retrieve contexts from the whole document pools. As mentioned in lines 161–180, "ToG-2 selectively explores entities neighboring the current topic entities on the KG, using the newly encountered entities to refine the retrieval scope, thereby enhancing both efficiency and accuracy." This refers to constraining the retrieval scope to the context of newly encountered entities rather than the entire document, thus improving retrieval efficiency.
>
> We agree that it would indeed be more efficient if retrieval were limited solely to the topic entity's context. However, the retrieval scope would be too narrow, likely leading to a significant drop in accuracy. As noted in the last paragraph of Section 3.1, before conducting graph-based retrieval, we first perform information retrieval within the topic entity's context and let the LLM determine whether the retrieved information is sufficient to answer the question. If not, we proceed with graph-based retrieval.
>
> Our experiments show that relying solely on the steps in Section 3.1 is often insufficient, as the information required to answer multi-hop reasoning questions is not confined to the topic entity's context. We will rephrase the relevant content in future versions to clarify our method more effectively.
>
> ### **Response to Question 3:**
> We realized that the legend colors for "LLM" and "BGE-Reranker" were mistakenly reversed in the figure. The actual experimental results align with our written analysis in the manuscript. Specifically, the performance of the BGE-Reranker improves as the number of entities increases, while the performance of the LLM decreases slightly. We sincerely apologize for this oversight in the figure and will correct it in the revised version of the manuscript. Thank you for catching this error and giving us the opportunity to clarify.
>
> ### **Response to Question 4:**
> Intuitively, we agree that the first-order neighbors of an entity in the query are often more important, as starting from a strong initial point can help avoid unnecessary exploration. In fact, our Topic Prune (TP) mechanism was designed with a similar consideration—it flexibly filters out less relevant topic entities based on the specific query. This can be seen as a simplified implementation of the dynamic approach you mentioned. Based on our observations, in most cases, the number of initial entities after TP is already smaller than the maximum exploration width W we set.
> We are also exploring and testing more dynamic graph traversal strategies, such as using the local information obtained from the current iteration to decide between breadth-first or depth-first search. Additionally, we are investigating how to implement such dynamic planning without incurring additional API costs. We appreciate your suggestion, as it aligns closely with some of the directions we are actively working on. Thank you!

---

> ### Author Response · Authors · 2024-11-18
> **Feedback to Official Review by Reviewer HRGs (Part II)**
>
> We appreciate the reviewer raising these questions and will rephrase the relevant content in future versions.

---

> ### Author Response · Authors · 2024-11-24
>
> Thank you for your valuable efforts, which have significantly contributed to improving the quality of the paper.
>
> We have updated the manuscript and would kindly inform you of the following updates based on your suggestions:
>
> Regarding "Weakness 2", the analysis of the hyperparameters for Threshold Setting in Relation Prune has been updated in Appendix B.5 (Impact of Threshold Setting in Relation Prune). Regarding "Weakness 3", we have corrected all the typos you mentioned. Regarding the Images (3b) and Tables (7, 16, 17) that caused unclearness mentioned in "Questions 1 & 3 & 5", we have made the necessary corrections. Thank you again for your patience and thoroughness!
>
> In addition, we have made further changes based on feedback from other reviewers. For more details, please refer to "General Feedback to All Reviewers" (https://openreview.net/forum?id=oFBu7qaZpS&noteId=1HmLBPo3eA)

---

> > ### Comment · Reviewer_HRGs · 2024-11-26
> >
> > Thank you for the detailed response. I would like to keep my scores.

---

### Official Review · Reviewer_wm5v · 2024-10-30

**Soundness:** 3
**Presentation:** 2
**Contribution:** 2
**Rating:** 6
**Confidence:** 5

**Summary:**

This paper introduces Think-on-Graph 2.0 (ToG-2), a novel hybrid retrieval-augmented generation (RAG) framework that aims to enhance large language model (LLM) reasoning by combining structured knowledge from knowledge graphs (KGs) with unstructured knowledge from texts. The key innovation is the "tight-coupling" approach between KG-based and text-based RAG, where KGs guide deeper context retrieval while textual context enables more precise graph retrieval. The framework iteratively performs knowledge-guided context retrieval and context-enhanced graph retrieval to obtain comprehensive information for LLM reasoning. The authors demonstrate ToG-2's effectiveness through extensive experiments across multiple knowledge-intensive datasets, showing state-of-the-art performance on 6 out of 7 benchmarks and the ability to elevate smaller LLMs' reasoning capabilities to be comparable with larger models.

**Strengths:**

1. Technical Innovation: The tight-coupling approach between KG and text-based RAG is novel and well-motivated, addressing limitations of previous hybrid approaches.

2. Comprehensive Evaluation: The experimental validation is thorough, covering multiple datasets and including detailed ablation studies. The creation of ToG-FinQA shows commitment to robust evaluation.

3. Implementation Details: The paper provides detailed implementation guidance and pseudocode, making it reproducible.

4. Analysis Depth: The manual analysis and runtime studies provide valuable insights into the framework's behavior and efficiency.

**Weaknesses:**

1. Scalability Analysis: While runtime comparisons are provided, there could be more discussion about the framework's scalability with very large knowledge graphs or document collections.

2. Error Analysis: Though there is some manual analysis of error cases, a more systematic categorization of failure modes could be valuable.

3. Hyperparameter Sensitivity: While some hyperparameters are studied (width and depth), a more comprehensive analysis of other parameters' impact would be helpful.

4. ToG-FinQA Dataset: While valuable, the dataset is relatively small (97 QA pairs). A larger dataset would strengthen the domain-specific evaluation.

**Questions:**

1. How does the performance of ToG-2 scale with significantly larger knowledge graphs or document collections?

2. Could you provide more details about the failure modes of the system and potential strategies to address them?

3. How sensitive is the system to the quality of the initial entity linking step? What happens when entity linking fails?

---

> ### Author Response · Authors · 2024-11-20
> **Feedback to Official Review by Reviewer wm5v (Part I)**
>
> **We sincerely thank the reviewer for their meticulous and insightful review, as well as the time and effort devoted to thoroughly evaluating our work.** Below, we address weaknesses, and questions raised, providing clarifications and discussing potential improvements.
>
> ### **Response to Weakness 1 & Question 1:**
>
> As with ToG, we use the March 2023 dump of Wikidata as the corresponding KG for the six datasets in Table 1.  We also follow the Wikidata deployment of ToG to make a fair comparison. The Wikidata dump we use contains 86,270,184 entities and 710,220,217 edges (each edge representing a relationship between two entities) after full deployment. This scale exceeds that of most KGs used in real-world scenarios.
>
> Additionally, the ToG-FinQA dataset we constructed includes a KG with 671,806 entities and 565,994 edges as reported in Table 8. We believe the experimental results on these two KG of different scales sufficiently demonstrate the scalability of our method.
>
> ### **Response to Weakness 2 & Question 2:**
>
> In Table 5, we provide a categorized analysis of errors, which we believe captures the majority of key cases encountered during the reasoning process, including both correct and incorrect outputs. To further enhance clarity, we would be happy to share some specific examples we observed:
>
> **Question**: One can watch Bugs Bunny on the television.
>
> **ToG2.0 Response**: {Yes}, Bugs Bunny can be watched on television. The references mention the history and success of the Bugs Bunny series, indicating its presence on various channels and streaming services over the years. The knowledge triplets also confirm Bugs Bunny's association with iconic works like *Looney Tunes* and notable creators like Tex Avery and Mel Blanc.
>
> **ToG Response**: {Yes}, the claim is {True}. Based on the knowledge triplets, Bugs Bunny is a beloved character from iconic works such as *Looney Tunes*, which was created by Warner Bros, and has been featured extensively in television broadcasts since its creation.
>
> **Label**: True
>
> **Error Reason**: The model failed to produce the expected output format {True}. The LLM did not strictly follow the instructions, which is a low-probability event but accounts for a large proportion of errors.
>
> ---
>
> **Question**: Quark served the man a drink on *Star Trek: Deep Space Nine*.
>
> **ToG2.0 Response**: {No}, the claim is {False}. The claim that "Quark served the man a drink on *Star Trek: Deep Space Nine*" cannot be evaluated based on the given information. The references provide information about the character Quark being a Ferengi bartender on *Deep Space Nine*, but there is no specific mention of him serving a drink to a man.
>
> **ToG Response**: {Yes}, the claim is {True}. The knowledge triplets confirm Quark's association with *Star Trek: Deep Space Nine* and his role as a bartender, serving drinks to patrons in the series.
>
> **Label**: Ture
>
> **Error Reason**: The model's failure stems from two main issues: overcautiousness and semantic ambiguity. Despite identifying the golden sentence, "Quark was the bartender on *Deep Space Nine*," the model's excessive caution prevents it from making implicit inferences without explicit evidence. Additionally, the semantic ambiguity in the question, particularly the use of "the man," implies a specific individual without sufficient context, leading to misinterpretation. We believe ToG, as it only provides information in the form of triples, has relatively sparse information density, and since triples are not natural text, the model needs to perform additional interpretation of them. Therefore, ToG is more likely to answer correctly in this case, ToG2.0’s detailed and cautious document analysis aligns with common human understanding, but it diverges from the ground truth, reflecting a mismatch between the dataset's annotations and nuanced inferences.

---

> ### Author Response · Authors · 2024-11-20
> **Feedback to Official Review by Reviewer wm5v (Part II)**
>
> ---
>
> **Question**: Parkinson's disease causes various hallucinations in people.
>
> **ToG2.0 Response**: {Yes}, the claim is {True}. The retrieved references mention that Parkinson's disease psychosis (PDP) is present in around 20 percent of cases and comprises hallucinations, illusions, and delusions. This confirms that Parkinson's disease can indeed cause various hallucinations in people.
>
> **ToG Response**: {No}, the claim is {False}. The retrieved references do not provide any evidence linking Parkinson's disease to causing various hallucinations in people. As no supporting information was found, it is concluded that this condition does not lead to hallucinations.
>
> **Label**: False
>
> **Error Reason**: As a non-expert, we might also believe that the LLM's basis and conclusion are correct—that Parkinson's disease can indeed cause hallucinations. However, the golden label is "False, Parkinson's has not been known to cause hallucinations." While we cannot assert whether the question itself is flawed, the judgment made by ToG2.0 aligns with that of an average human. Since ToG2.0 can access richer details in the documents, it is able to make the above reasonable inference. In contrast, the LLM in ToG cannot find effective information from the triples and makes a reckless assertion.
>
> Conclusion:
>
> While ToG excels in scenarios demanding rapid, triplet-based reasoning, ToG2.0 demonstrates its strength in cases requiring detailed contextual reasoning, albeit at the cost of overcautiousness or misalignment with less explicit ground truths. On the other hand, ToG2.0's cautious behavior tendency can help prevent the generation of hallucinations and the propagation of errors. Additionally, the quality issues in some questions within the dataset also limit its upper bound of performance.
>
> > ... potential strategies to address them.
>
> Regarding the cases summarized in Table 5, we are actively exploring and testing strategies to address these issues:
> 1. **Hallucination**: Compared to traditional RAG approaches, our tight-coupling paradigm has already reduced hallucination to a very low probability. For applications with stricter requirements on hallucination tolerance, such as in finance or healthcare, our method can easily incorporate advanced fact-checking modules to further eliminate hallucination risks.
> 2. **No Info**: This often arises when the corpus lacks relevant information. A potential solution is to leverage the clues obtained through graph exploration in ToG-2 and integrate them with web search agents to gather additional knowledge from diverse sources. Additionally, the choice of graph structures could be enhanced by adopting graph construction methods such as GraphRAG [1] or LightRAG [2].
> 3. **False Negative**: These cases can potentially be addressed by employing evaluation mechanisms based on ChatGPT or open-source LLMs to implement more flexible assessments. However, we acknowledge that such an approach may introduce significant computational and efficiency overhead, which remains a common challenge for many researchers.
>
> ### **Response to Question 3:**
>
> For datasets like WebQSP, which come with manually annotated high-quality topic entities, we directly utilize the topic entities provided by the dataset. For datasets like HotpotQA that do not include topic entities, we use the Azure API for named entity recognition (NER) as a convenient option, as mentioned in Lines 213-215 of the paper. Additionally, other open-source NER tools, such as Stanford CoreNLP or HanLP, can also be employed. These open-source methods have demonstrated strong performance in entity recognition and disambiguation, particularly for identifying key entities in QA tasks, which are often proper nouns or domain-specific terms, making them relatively straightforward to detect.
>
> In our study, we observed that the notable issue is not the failure to extract topic entities but rather the extraction of too many entities, which can introduce noise into downstream processes. To address this, we employ Topic Prune (TP), which effectively prunes irrelevant entities and ensures that only entities closely aligned with the question's intent are retained. For instance, after applying TP, the average number of entities retained for HotpotQA questions is 1.36, which aligns closely with human observations of the dataset, where most questions involve one or two key entities.

---

> ### Author Response · Authors · 2024-11-20
> **Feedback to Official Review by Reviewer wm5v (Part III)**
>
> ### **Response to Weakness 3:**
>
> We agree that understanding the impact of various hyperparameters is essential for evaluating the robustness and adaptability of the system. Beyond the parameters we have studied, such as width $W$, depth $D$, and context length, we conducted an additional hyper-parameter experiment for thresholds setting in relation pruning.
>
> Specifically, we compared the effects of three thresholds (0.2, 0.5, and 0.8) on the final results in HotpotQA. The **Avg. retained score** represents the average score of all relations that pass through the thresholds filtering. The results are as follows:
>
> | Thresholds | Results | Avg. retained score |
> | --- | --- | --- |
> | 0.2 | 42.9% | 0.65 |
> | 0.5 | 43.2% | 0.68 |
> | 0.8 | 40.0% | 0.92 |
>
> When the threshold is set between 0.2 and 0.5, its impact on the final result is minimal. However, when the threshold increases to 0.7, it significantly weakens the results. We attribute this to two main reasons:
>
> 1. During the relation pruning stage, the relations passing the threshold are further filtered by selecting the top relations based on the width $W$. Therefore, even if the threshold is set very low, relations with extremely low scores will ultimately fail to be included in the Top W. This explains why the **Avg. retained score** is very close when the threshold is 0.2 or 0.5. However, setting a reasonable threshold is still necessary to prevent the inclusion of entirely irrelevant relations in cases where very few relations have found and are scored low (e.g., as low as 0.1).
> 2. Relations with moderate relevance scores (e.g., 0.3–0.7) can potentially lead to useful information. A lower threshold allows for more tolerance for such potential. When the threshold is raised to 0.7, it overly amplifies the impact of LLM’s judgement during relation pruning on the entire process, leading to the loss of potentially valuable relations.
>
> **We once again thank the reviewer for their valuable feedback and thoughtful suggestions, which have helped us improve our work. If there are any further questions or concerns, we would be happy to address them.**

---

> ### Author Response · Authors · 2024-11-24
>
> Thank you for your valuable efforts, which have significantly contributed to improving the quality of the paper.
>
> We have updated the manuscript and would kindly inform you of the following updates based on your suggestions:
>
> Regarding "Weakness 2 & Question 2", the new case studies with detailed bad cases have been updated in Appendix B.2 (Supplementary Analysis for the Main Results). Regarding "Weakness 3", the analysis of the hyperparameters for Threshold Setting in Relation Prune has been updated in Appendix B.5 (Impact of Threshold Setting in Relation Prune).
>
> In addition, we have made further changes based on feedback from other reviewers. For more details, please refer to "General Feedback to All Reviewers" (https://openreview.net/forum?id=oFBu7qaZpS&noteId=1HmLBPo3eA)

---

> ### Comment · Reviewer_wm5v · 2024-11-27
> **Seeking a complete code execution guide**
>
> Thank you for your detailed response. I have thoroughly re-read your paper and would like to confirm some technical implementations in your code. However, I noticed that the README file is empty, and I’m unsure how to run the code. Could you please update the README to include a comprehensive guide on how to run your code?
>
> As reproducibility is the most important factor for such work, I will temporarily lower the rating to 3 until this is addressed.

---

> > ### Author Response · Authors · 2024-11-27
> >
> > Dear Reviewer,
> >
> > Thank you very much for bringing this issue to our attention. We truly appreciate your meticulous review and feedback.
> >
> > 1. We would like to clarify that after submitting the manuscript, we have made improvements to our github repository to enhance its clarity. However, we just realized that the updated code and README were not automatically synchronized to the anonymous repository. We sincerely apologize for this oversight and are very grateful for your reminder. We have now updated the anonymous repository with README file includes a detailed guide on how to run our experiments.
> > 2. If you encounter any further issues while reviewing the code, please feel free to raise them. We are committed to addressing all your concerns and ensuring the reproducibility of our work. We also kindly request you to reconsider the rating after these updates.
> >
> > Once again, thank you for your valuable feedback and understanding.
> >
> > Best regards,

---

> > > ### Comment · Reviewer_wm5v · 2024-11-27
> > >
> > > Thank you for providing the instructions!
> > >
> > > I will get the code thoroughly checked and get back to you soon.
> > >
> > > Best regards,
> > >
> > > The Reviewer

---

> > > ### Comment · Reviewer_wm5v · 2024-11-27
> > >
> > > I have checked the code and changed the rating back to 6. Thanks.

---

### Official Review · Reviewer_8nFX · 2024-11-02

**Soundness:** 3
**Presentation:** 3
**Contribution:** 2
**Rating:** 6
**Confidence:** 5

**Summary:**

This manuscript presents a framework designed to augment the reasoning capabilities of Large Language Models (LLMs) through the integration of knowledge graphs. The framework addresses several limitations inherent in current methodologies by implementing three key strategies: hybrid knowledge exploration, knowledge-guided graph search, and knowledge-guided context retrieval. The objective is to mitigate issues such as constrained knowledge retrieval and the scarcity of informed knowledge within LLMs.

**Strengths:**

- The manuscript adeptly delineates the constraints of LLMs in reasoning through complex queries and proposes a systematic and coherent methodology to surmount these limitations.

- The incorporation of hybrid knowledge, derived from both knowledge graphs and wiki documents, constitutes an intuitive approach to enhancing the inferential competencies of LLMs, which, to the best of my knowledge, is a novel contribution to the field.

- Table 1 offers a comparative analysis with appropriately set benchmarks, demonstrating the superiority of the proposed ToG 2.0 model as a new SOTA solution. The authors further substantiate this claim through empirical experiments.

- The figures and graphics were well done and helped with understanding the methodology of the paper substantially.

**Weaknesses:**

- The incremental contributions of ToG 2.0 over its predecessor, ToG, are noteworthy but not transformative.
> Specifically, the core improvements include the addition of context-retrieval and context-guided entity pruning to ToG's existing Relation exploration and Entity exploration.


- Although  LLM costs is not normally a point of significant concern for the soundness/rigor of a work. However, the methodology is financially demanding, as processes such as Knowledge Exploration, Relation Pruning, Relation Discovery, Entity Pruning, and Reasoning all necessitate frequent calls to the LLM API. The associated high costs may deter further exploration by researchers within the Graph RAG community.
> Constructively, it is suggested that the authors elaborate on a refined cost analysis and contemplate the implementation of optimizations to reduce API calls. Such measures would effectively address the raised concerns about `extensive utilization of LLM APIs’ and simultaneously provide pragmatic suggestions for further enhancement.


- I harbor reservations regarding the efficiency of this approach. Empirically, processes that require extensive utilization of LLM APIs are inherently time-consuming. As evidenced by the time expenditures for Relation Pruning (RP), Entity Pruning (EP), and Reasoning detailed in Table 3, the task at hand is laborious. The documentation lacks clarity on whether Table 7 provides the RP, EP, and reasoning times for each individual sample. If it does, then for multi-hop Knowledge Graph (KG) reasoning tasks, compounded by the size of the dataset, the task becomes significantly time-consuming when juxtaposed with direct LLM-only methods or Text-based RAG approaches. Additionally, ToG 2.0 necessitates grounding in the vast corpus of external Wikipedia data.
> I contend that evaluating the efficiency of this method solely in comparison to ToG 1.0 lacks impartiality.

**Questions:**

- What about total runtime and API call counts compared to other baselines like direct LLM methods and text-based RAG approaches ?
- Table 7 should clarify whether it reports the reasoning time for each discrete query or for each RP and EP iteration, given that multi-hop reasoning entails successive RP and EP cycles.

- The question also arises: Why does ToG outperform ToG 2.0 in the Creak dataset? To make it more actionable, more detailed analysis of the performance on the Creak dataset should be provided, including potential reasons for this discrepancy and its implications for the method's strengths and limitations.

---

> ### Author Response · Authors · 2024-11-16
> **Feedback to Official Review of Reviewer 8nFX (Part I)**
>
> **Thank you for your insightful feedback and valuable suggestions. We sincerely appreciate your conscientious comments regarding our work. Below, we provide detailed analysis and explanation to address your concerns:**
>
>
> ### **Response to Weakness 2 \& Question 1:**
> **1. Comparison of API Call Counts and Runtime**: To better illustrate the performance of our method compared to the baselines, we have analyzed the average total runtime and API call counts, as shown in the table below:
> | Dataset       | Method               | Avg Total Time | API Calls |
> |---------------|----------------------|----------------|-----------|
> | AdvHotpotQA   | ToG2  | 27.3s          | 5.4       |
> |               | NaiveRAG            | 10.2s          | 1         |
> |               | ToG                 | 69.3s          | 16.3      |
> |               | CoK                 | 30.1s          | 11        |
> - **ToG**: For each question, ToG requires up to $2W \cdot D + D + 1$ API calls, where $W$ is the maximum width and $D$ is the maximum iteration depth. This structure enables comprehensive multi-hop reasoning and knowledge exploration.
> - **CoK**: To ensure fairness, we restricted the knowledge domain to factual data from WikiData, Wikipedia, and DPR. Additionally, we adopted the default `update_rationales_at_once` setting from the original implementation. CoK has a fixed API call count of 11 per question: $1$ domain selection, $1$ chain-of-thought sub-question generation, $2 \times 3$ knowledge retrieval calls, $2$ rationale editing calls, and $1$ final answer generation.
> - **NaiveRAG**: To simulated a retrieval corpus, we use all entity documents encountered during ToG2.0 iterations. We utilized a two-stage retrieval system combining BM25 and BGE-Reranker.
> - **ToG2.0**: For each question, ToG2 requires up to $2D + (D-1) + 1$ API calls. Specifically:
>     - $2D$: Calls for Relation Pruning (RP) and Reasoning during $D$ iterations.
>     - $D-1$: Query rewriting between iterations.
>     - $1$: Topic pruning (TP).
> **2. Performance Advantages**
>
> Compared to the baselines, **ToG2.0** achieves a good balance between cost-efficiency and performance. It reduces the total API calls and runtime significantly while maintaining robust reasoning capabilities.
>
> ---
>
> **3. Future Optimization Directions**
>
> To further address concerns regarding the cost and runtime of LLM API calls, we plan to:
>
> - **Leverage lightweight language models**: We will explore fine-tuned smaller models for query rewriting and topic pruning to reduce reliance on expensive LLM APIs.
> - **Optimize API call structure**: By introducing dynamic programming-based techniques, we aim to minimize redundant calls and further enhance the efficiency of our method.
>
> ### **Response to Weakness 3 \& Question 2:**
> Table 7 provides the average number of iterations in answering each discrete question, along with the average runtime for each iteration, specifically breaking down into the RP (Relation Pruning), EP (Entity Pruning), and Reasoning steps. Notably, in our default setup, RP utilizes a relation prune combination approach, and each iteration involves making only two API calls: one for RP and one for Reasoning. EP is vector-based and does not require API calls.

---

> ### Author Response · Authors · 2024-11-16
> **Feedback to Official Review of Reviewer 8nFX (Part II)**
>
> ### **Response to Question 3:**
> To address the reviewer’s insightful question Q3,
>
> We manually analyzed the reasoning process for 50 examples from the Creak dataset to better understand the performance differences between ToG and ToG2.0, following the format of Table 4 in our paper:
>
> | Answer Type |  ToG2.0 |
> | --- | --- |
> | Triple-enhanced Answer | 42% |
> | Doc-enhanced Answer | 24% |
> | Both-enhanced Answer | 34% |
>
> Due to the relatively simple nature of the questions in Creak, results can be inferred directly using triple. Therefore, ToG has already performed well on Creak, and the richer entity document information utilized by ToG 2.0 can not further provide significant additional benefits on this dataset. As shown in the table above, compared to HotpotQA, ToG 2.0 produces more Triple-enhanced Answers on Creak.
>
> We also conducted a detailed case study of the behavior of ToG and ToG2.0 on the Creak dataset. Our investigation highlights three representative cases that elucidate the primary reasons for the observed performance discrepancy.
>
> **Question**: One can watch Bugs Bunny on the television.
>
> **ToG2.0 Response**: {Yes}, Bugs Bunny can be watched on television. The references mention the history and success of the Bugs Bunny series, indicating its presence on various channels and streaming services over the years. The knowledge triplets also confirm Bugs Bunny's association with iconic works like *Looney Tunes* and notable creators like Tex Avery and Mel Blanc.
>
> **ToG Response**: {Yes}, the claim is {True}. Based on the knowledge triplets, Bugs Bunny is a beloved character from iconic works such as *Looney Tunes*, which was created by Warner Bros, and has been featured extensively in television broadcasts since its creation.
>
> **Label**: True
>
> **Error Reason**: The model failed to produce the expected output format {True}. The LLM did not strictly follow the instructions, which is a low-probability event but accounts for a large proportion of errors.
>
> ---
>
> **Question**: Quark served the man a drink on *Star Trek: Deep Space Nine*.
>
> **ToG2.0 Response**: {No}, the claim is {False}. The claim that "Quark served the man a drink on *Star Trek: Deep Space Nine*" cannot be evaluated based on the given information. The references provide information about the character Quark being a Ferengi bartender on *Deep Space Nine*, but there is no specific mention of him serving a drink to a man.
>
> **ToG Response**: {Yes}, the claim is {True}. The knowledge triplets confirm Quark's association with *Star Trek: Deep Space Nine* and his role as a bartender, serving drinks to patrons in the series.
>
> **Label**: Ture
>
> **Error Reason**: The model's failure stems from two main issues: overcautiousness and semantic ambiguity. Despite identifying the golden sentence, "Quark was the bartender on *Deep Space Nine*," the model's excessive caution prevents it from making implicit inferences without explicit evidence. Additionally, the semantic ambiguity in the question, particularly the use of "the man," implies a specific individual without sufficient context, leading to misinterpretation. We believe ToG, as it only provides information in the form of triples, has relatively sparse information density, and since triples are not natural text, the model needs to perform additional interpretation of them. Therefore, ToG is more likely to answer correctly in this case, ToG2.0’s detailed and cautious document analysis aligns with common human understanding, but it diverges from the ground truth, reflecting a mismatch between the dataset's annotations and nuanced inferences.
>
> ---
>
> **Question**: Parkinson's disease causes various hallucinations in people.
>
> **ToG2.0 Response**: {Yes}, the claim is {True}. The retrieved references mention that Parkinson's disease psychosis (PDP) is present in around 20 percent of cases and comprises hallucinations, illusions, and delusions. This confirms that Parkinson's disease can indeed cause various hallucinations in people.
>
> **ToG Response**: {No}, the claim is {False}. The retrieved references do not provide any evidence linking Parkinson's disease to causing various hallucinations in people. As no supporting information was found, it is concluded that this condition does not lead to hallucinations.
>
> **Label**: False
>
> **Error Reason**: As a non-expert, we might also believe that the LLM's basis and conclusion are correct—that Parkinson's disease can indeed cause hallucinations. However, the golden label is "False, Parkinson's has not been known to cause hallucinations." While we cannot assert whether the question itself is flawed, the judgment made by ToG2.0 aligns with that of an average human. Since ToG2.0 can access richer details in the documents, it is able to make the above reasonable inference. In contrast, the LLM in ToG cannot find effective information from the triples and makes a reckless assertion.

---

> ### Author Response · Authors · 2024-11-16
> **Feedback to Official Review of Reviewer 8nFX (Part III)**
>
> **Conclusion**:
> While ToG excels in scenarios demanding rapid, triplet-based reasoning, ToG2.0 demonstrates its strength in cases requiring detailed contextual reasoning, albeit at the cost of overcautiousness or misalignment with less explicit ground truths. On the other hand, ToG2.0's cautious behavior tendency can help prevent the generation of hallucinations and the propagation of errors. Additionally, the quality issues in some questions within the dataset also limit its upper bound of performance.
>
> **Thank you for your time and thoughtful review. Please let us know if further clarification or elaboration is needed.**

---

> ### Author Response · Authors · 2024-11-18
> **Feedback to Official Review of Reviewer 8nFX (Part IV)**
>
> ### **Response to Weakness 1:**
>
> ToG is a KG-based RAG framework designed for reasoning tasks over KGs, such as KGQA and KG-based fact check, relying solely on KG triples and unable to leverage text corpora, a common knowledge source for LLMs.
>
> ToG 2.0, however, is a hybrid RAG approach that integrates structured KG knowledge with unstructured text corpora to enhancing LLM reasoning. Experiments show that ToG struggles with text-based RAG datasets like AdvHotpotQA and ToG-FinQA, which require detailed text retrieval, while ToG 2.0 excels in these cases.
>
> Text-based RAG is more common in real-world applications than KG-based RAG due to the difficulty and information loss in KG construction. From this perspective, ToG 2.0 offers significantly better generalization capabilities than ToG, as it can utilize documents as external knowledge sources and is less dependent on the KG’s quality (ToG-FinQA, for instance, uses a small KG with only seven relation types). ToG 2.0 also ranks entities efficiently via embedding similarity rather than LLM ranking which is used by ToG.
>
> In summary, ToG is a KG-based RAG approach, while ToG 2.0 is a hybrid RAG appraoch, showing improved performance, efficiency, and generalization across both various knowledge-intensive reasoning tasks.

---

> ### Comment · Reviewer_8nFX · 2024-11-22
>
> - Thank you for your reply. The PART I responses have addressed my concern. It is recommended that the 'Comparison of API Call Counts and Runtime' be included in the final version.
>
> - PART II, III, and IV have provided too much clarification, which requires time to carefully review. I will try to finish reading and reply to you promptly.

---

> > ### Author Response · Authors · 2024-11-22
> > **Feedback from authors**
> >
> > Thank you for your suggestions and for acknowledging our response. If you have any further questions, we would be happy to provide additional clarifications. If you think our response has addressed your concerns, we sincerely hope you might kindly consider revising your score. Thank you once again.

---

> > > ### Comment · Reviewer_8nFX · 2024-11-24
> > >
> > > Thank you for your thoughtful response, which addresses most of my concerns.
> > > To fully evaluate the extensions during the rebuttal process, a revised manuscript is required and I will retain my score for the submitted manuscript.

---

> ### Author Response · Authors · 2024-11-24
>
> Thank you for your valuable efforts, which have significantly contributed to improving the quality of the paper.
>
> We have updated the manuscript and would kindly inform you of the following updates based on your suggestions:
>
> The clarification for "Weakness 3 & Question 2" and the additional experiment addressing your concern in "Weakness 2 & Question 1" regarding runtime and API calls, titled "Comparison of API Call Counts and Runtime Across Methods," has been updated in Appendix B.4 (Runtime Analysis Results).
>
> In response to your concern in "Question 3," the analysis of the discrepancy between ToG and ToG-2 on the Creak dataset and the case studies, has been updated in Appendix B.2 (Supplementary Analysis for the Main Results).
>
> In addition, we have made further changes based on feedback from other reviewers. For more details, please refer to "General Feedback to All Reviewers" (https://openreview.net/forum?id=oFBu7qaZpS&noteId=1HmLBPo3eA)

---

> > ### Comment · Reviewer_8nFX · 2024-11-25
> > **Update my score**
> >
> > After a thorough review, I have decided to revise my score upward.
> > Good Luck！

---

### Official Review · Reviewer_9CdM · 2024-11-04

**Soundness:** 4
**Presentation:** 3
**Contribution:** 3
**Rating:** 8
**Confidence:** 3

**Summary:**

This paper presents Think-on-Graph 2.0 (ToG-2), a RAG framework that integrates both unstructured text and structured knowledge graphs in a tightly coupled manner to improve retrieval quality and reasoning accuracy for LLMs. ToG-2 addresses key limitations of current RAG methods, which often fail to retrieve sufficiently deep and contextually rich information for complex reasoning tasks. By alternating between context retrieval and KG-based retrieval, ToG-2 effectively links and enriches information sources, guiding LLMs in producing more reliable and contextually informed answers. On various tasks (multi-hop KBQA, multi-hop complex document QA, slot filling, and fact verification), ToG-2 achieves satisfying performance in comparison with different RAG baselines (text-based, KG-based, and hybrid).

**Strengths:**

+ The proposed tight-coupling approach is well-motivated. It uses KGs to guide the retrieval of document contexts while simultaneously treating these documents as entities in the KG, thereby creating a feedback loop that enhances both retrieval depth and context fidelity.

+ The iterative retrieval process is intuitive. It mimics human reasoning by continuously refining the search for information based on initial retrieval results.

+ Experiments are comprehensive. Six datasets across four different tasks are examined. Baselines using CoT, text-only RAG, KG-based RAG, and hybrid RAG are compared. Besides performance comparison, the authors also conduct ablation studies, efficiency analysis, and case studies to verify some of their design choices.

**Weaknesses:**

- Statistical significance tests are not conducted. It is not clear whether the gaps between ToG-2 and the baselines are statistically significant or not in Table 1. In fact, the improvement is subtle in some columns. Also, the statistical significance of ablation studies should be considered.

- The quality and completeness of the KG used in ToG-2 can significantly influence its retrieval accuracy and reasoning performance. The effect of KG quality on the performance of ToG-2 is not accurately quantified. Although the authors add a ToG-FinQA dataset in their experiments with a newly built KG, the quality of this KG is also unknown. It would be valuable to explore the performance of ToG-2 with incomplete WikiData.

**Questions:**

- Could you conduct a statistical significance test (e.g., two-tailed t-test) to compare ToG-2 with the baselines in Table 1, and report the p-values?

- Could you study the effect of KG quality on the performance of ToG-2 by using incomplete WikiData as KG (e.g., with 10%, 20%, or 50% of the entities/relations)?

---

> ### Author Response · Authors · 2024-11-20
> **Feedback to Official Review of Reviewer 9CdM (Part I)**
>
> **We sincerely appreciate the reviewer‘s time and effort in providing thoughtful feedback, as well as the recognition and attention to our work. Regarding the concern raised, we would like to provide further clarification:**
>
> ### **Response to Weakness 1 & Question 1:**
>
> In prior RAG research, including **Think-on-Graph** and **Chain-of-Knowledge**, the statistical significance of differences between models and baselines was not reported in experimental tables. Following this precedent, we did not include p-values in the current version of our work. Furthermore, our study adopted a deterministic setup by setting the temperature to 0 for all generation processes, ensuring reproducibility and minimizing variability in the outputs of the LLM components. This approach reduced the need for multiple runs, which typically underpin significance tests.
>
> That said, we fully agree that statistical significance is an important aspect of evaluating model performance. To address this, we have computed pairwise t-test ****for select datasets where we had access to both ToG-2 and baseline results. However, as many of the baseline results in Table 1 were directly cited from their respective original papers or other related works, we do not have access to the detailed outputs of these baseline models for each individual question. Consequently, we are unable to test for these baselines at this time.
>
> On **WebQSP, Zero-Shot-RE,QALD-10-en and Creak,** we computed the **pairwise t-test** for ToG-2 versus CoK, ToG as follows:
>
> | **(ToG-2 vs. )Method** | **WebQSP** | **Zero-Shot-RE** | **QALD-10-en** | **Creak** |
> | --- | --- | --- | --- | --- |
> | Cok | p < 0.05 | p < 0.01 | p < 0.05 | p < 0.05 |
> | ToG | p < 0.05 | p < 0.05 | p < 0.05 | p > 0.9 |
>
> ToG-2 significantly outperforms CoK and ToG on most datasets (p < 0.05), except on Creak where the difference with ToG is not significant (p > 0.9), indicating similar performance on this dataset. Moving forward, we plan to re-test all baseline models to generate their outputs for each question, enabling a direct comparison with ToG-2 and the calculation of p-values across all datasets. These updates will be included in future versions of our work.
>
> ### **Response to Weakness 2 & Question 2:**
>
> To investigate the impact of incomplete knowledge graphs (KGs) on ToG-2.0, we conducted experiments on a subset of randomly sampled HotpotQA questions (100 hops in total) using simulated incomplete KGs. The incompleteness was introduced by randomly discarding relations and entities discovered during the Relation Discovery (Equation 1) and Entity Discovery (Equation 4) stages. Specifically, we tested scenarios where 30%, 50%, and 80% of the discovered entities and relations were retained, while the rest were randomly removed. This systematic evaluation provided insights into the model's adaptability to varying levels of KG completeness.
>
> ### Results:
>
> | KG Completeness (%) | Exploration Strategy | EM (%) |
> | --- | --- | --- |
> | 100% | Default ($W = 3$, $D = 3$) | 43 |
> | 80% | Default | 41 |
> | 50% | Default | 35 |
> | 30% | Default | 23 |
> | 30% | Adjusted ($W = 8$, $D = 2$) | 29 |
> 1. **80% Completeness:**
>
>     ToG-2.0 maintained strong performance with minimal impact compared to the fully complete KG, demonstrating its robustness at high completeness levels.
>
> 2. **50% Completeness:**
>
>     We observed moderate performance degradation, indicating that ToG-2.0 is resilient to moderate KG incompleteness.
>
> 3. **30% Completeness:**
>
>     Performance dropped significantly due to sparse connections in the graph. To mitigate this issue, we tried to adjust the exploration strategy during reasoning:
>
>     - **Exploration Width ($W$):** Increased to 8.
>     - **Exploration Depth ($D$):** Reduced to 2.
>
>     These adjustments allowed the model to cover a broader search scope, compensating for the missing information. As a result, we observed a significant recovery in performance, bringing it closer to an acceptable level despite the severe KG incompleteness.
>
>
> ### Conclusion:
>
> These findings highlight the adaptability of ToG-2.0 under varying KG completeness levels and demonstrate potential strategies to mitigate the adverse effects of incomplete KGs. However, due to time constraints, our tests were conducted on a subset of 100 randomly sampled HotpotQA questions. Further experiments on a larger scale would provide a more comprehensive understanding of the model's performance in real-world scenarios.
>
> **Once again, we thank you for your thoughtful and constructive suggestion. Your rigorous attitude towards scientific research is highly commendable, and we are grateful for the opportunity to refine our work with your guidance.**

---

> ### Author Response · Authors · 2024-11-24
>
> Thank you for your valuable efforts, which have significantly contributed to improving the quality of the paper.
>
> We have updated the manuscript and would kindly inform you of the following updates based on your suggestions:
>
> Regarding our response to "Weakness 1 & Question 1",  we updated Appendix B.8 (Significance Test). For the experiment and analysis in response to "Weakness 2 & Question 2", we updated Appendix B.7 (Impact of Graph Completeness).
>
> In addition, we have made further changes based on feedback from other reviewers. For more details, please refer to "General Feedback to All Reviewers" (https://openreview.net/forum?id=oFBu7qaZpS&noteId=1HmLBPo3eA)

---

### Author Response · Authors · 2024-11-24
**General Feedback to All Reviewers**

**I would like to express my sincere gratitude to all reviewers for their time, insightful comments, and valuable suggestions, which have greatly contributed to improving the quality of this manuscript.** Based on all reviewers' comments, I have made several revisions to enhance the clarity, accuracy, and overall coherence of the manuscript.

**Below, I outline the key updates in the latest manuscript:**
* Additional analysis for the discrepancy between ToG and ToG-$2$ on the Creak dataset in Table 1 and case study have been updated in **Appendix B.2 (Supplementary Analysis for The Main Results)**.
* An additional experimental analysis, 'Comparison of API Call Counts and Runtime Across Methods,' has been updated in **Appendix B.4 (Runtime Analysis Results)**.
* An analysis of the hyperparameters for Threshold Setting in Relation Prune has been updated in **Appendix B.5 (Impact of Threshold Setting in Relation Prune)**.
* An experiment on the impact of Knowledge Graph (KG) incompleteness on the model's performance has been updated in **Appendix B.7 (Impact of Graph Completeness)**.
* The significance test has been updated in **Appendix B.8 (Significance Test)**.
* Some typographical errors in the images and text are corrected.
* Some clarifications for details.

(The distribution and layout of the overall content will be updated after careful arrangement in the future.)

---

### Meta-Review · Area_Chair_DRdN · 2024-12-20

**Metareview:**

This paper introduces Think-on-Graph 2.0 (ToG-2), a novel hybrid retrieval-augmented generation (RAG) framework designed to enhance large language model (LLM) reasoning by integrating structured knowledge from knowledge graphs (KGs) with unstructured knowledge from texts. The proposed tight-coupling approach is well-justified, leveraging knowledge graphs (KGs) to guide document context retrieval while integrating these documents as entities within the KG, forming a feedback loop that enhances both retrieval depth and context precision. Comprehensive experiments are conducted across six datasets and four tasks, comparing the framework with baselines including CoT, text-only RAG, KG-based RAG, and hybrid RAG. Additional ablation studies, efficiency analysis, and case studies further validate the design choices.

However, the contribution of ToG 2.0 over ToG is  incremental, primarily involving the addition of context-retrieval and context-guided entity pruning to the original framework's relation and entity exploration. The methodology is resource-intensive, requiring frequent LLM API calls for processes such as Knowledge Exploration, Relation Pruning, and Reasoning, which may deter researchers in the Graph RAG community due to high costs.

Constructive suggestions include providing a detailed cost analysis and exploring optimizations to reduce API usage, addressing concerns about efficiency and practicality.  All reviewers  agree that this paper should be accepted.

**Additional Comments On Reviewer Discussion:**

During the discussion phase, all reviewers agreed with the authors' responses, resulting in improved scores.

---

### Decision · Program_Chairs · 2025-01-22

Accept (Poster)